# TOWARDS THE MEMORIZATION EFFECT OF NEURAL NETWORKS IN ADVERSARIAL TRAINING

## ABSTRACT

Recent studies suggest that "memorization" is one necessary factor for overparameterized deep neural networks (DNNs) to achieve optimal performance. Specifically, the perfectly fitted DNNs can memorize the labels of many atypical samples, generalize their memorization to correctly classify test atypical samples and enjoy better test performance. While, DNNs which are optimized via adversarial training algorithms can also achieve perfect training performance by memorizing the labels of atypical samples, as well as the adversarially perturbed atypical samples. However, adversarially trained models always suffer from poor generalization, with both relatively low clean accuracy and robustness on the test set. In this work, we study the effect of memorization in adversarial trained DNNs and disclose two important findings: **(a)** Memorizing atypical samples is only effective to improve DNN's accuracy on clean atypical samples, but hardly improve their adversarial robustness and **(b)** Memorizing certain atypical samples will even hurt the DNN's performance on typical samples. Based on these two findings, we propose *Benign Adversarial Training (BAT)* which can facilitate adversarial training to avoid fitting "harmful" atypical samples and fit as more "benign" atypical samples as possible. In our experiments, we validate the effectiveness of BAT, and show that it can achieve better clean accuracy vs. robustness trade-off than baseline methods, in benchmark datasets such as CIFAR100 and Tiny ImageNet.

## 1 INTRODUCTION

It is evident from recent studies that the memorization effect (or benign overfitting) (Feldman, 2020; Feldman & Zhang, 2020; Bartlett et al., 2020; Chatterji & Long, 2020; Muthukumar et al., 2020) is one necessary factor for overparametrized deep neural networks (DNNs) to achieve the close-to-optimal generalization error. From the empirical perspective, the works (Feldman, 2020; Feldman & Zhang, 2020) suggest that modern benchmark datasets, such as CIFAR10, CIFAR100 (Krizhevsky et al., 2009) and ImageNet (Krizhevsky et al., 2012), always have very diverse data distributions, especially containing a large fraction of "atypical" samples. These atypical samples are both visually and statistically very different from other samples in their labeled class. For example, the images in the class "bird" may have a variety of sub-populations or species, with many (typical) samples in the main sub-population and other (atypical) samples in less-frequent and distinct sub-populations. Since these atypical samples are deviated from the main sub-population, DNNs can only fit these atypical samples by "memorizing" their labels. While, memorizing / fitting these atypical samples will not hurt the model performance on typical samples, but can boost DNNs' accuracy by correctly classifying the atypical samples appearing in the test set.

Similar to the classification models which are trained via empirical risk minimization (ERM) algorithms, adversarial training methods (Madry et al., 2017; Kurakin et al., 2016) are also devised to fit the whole training dataset. Specifically, adversarial training minimizes the model's error against adversarial perturbations (Goodfellow et al., 2014; Szegedy et al., 2013) by fitting the model on manually generated adversarial examples of all training data. Although adversarial training can fit and memorize all training data as well as their adversarially perturbed counterparts, they always suffer from both poor clean accuracy and adversarial accuracy (or robustness)[1] on the test set (Tsipras et al.,

---

[1]Clean Accuracy: Model's accuracy on unperturbed samples. Adversarial accuracy (robustness): Model's accuracy on adversarially perturbed samples. Without loss of generality, this paper discusses the adversarial accuracy under $l_\infty$-8/255 PGD attack (Madry et al., 2017).

2018; Schmidt et al., 2018). Recent study (Rice et al., 2020) indicates that, during the adversarial training process, one model's test adversarial accuracy will even keep decreasing as it hits higher training adversarial accuracy (during the fine-tuning epochs). Thus, it is natural to ask a question: *What is the effect of memorization in adversarial training? In particular, can memorizing the atypical samples and their adversarial counterparts benefit the model's accuracy and robustness?*

To answer this question, we first conduct preliminary studies to explore whether memorizing atypical samples in adversarial training can benefit the DNNs' test performance, especially on those test atypical samples. In Section 3.1, we implement PGD adversarial training (Madry et al., 2017) on CIFAR100 under WideResNet models (He et al., 2016) and fine-tune them until achieving the optimal training performance. From the results in Section 3.1, we observe that *the memorization in adversarial training can only benefit the clean accuracy of test atypical samples.* When the DNNs gradually fit/memorize more atypical samples, they can finally achieve fair clean accuracy close to $\sim 40\%$ on the test atypical set. However, the adversarial accuracy on test atypical set is constantly low ($\sim 10\%$) during the whole training process, even though the models can fit almost all atypical (adversarial) training samples. Based on the theoretical study (Schmidt et al., 2018), the adversarial robustness is hard to generalize especially when the training data size is limited. Since every single atypical sample is distinct from the main sub-population and rarely appears in the training set, the data complexity for each specific atypical sample is very low. Thus, its adversarial robustness can be extremely hard to generalize. Notably, for datasets such as CIFAR100, the entire atypical set covers at least $\sim 40\%$ samples of the whole dataset. Therefore, completely failing on atypical samples could be one of the key reasons leading to the poor robustness generalization of DNNs.

Furthermore, we find that *in adversarial training, fitting atypical samples will even hurt DNNs' performance on those "typical" samples (the samples in the main sub-population).* In the Section 3.2, we again implement PGD adversarial training (Madry et al., 2017) on CIFAR100 for several trails, which are trained with different amount of atypical samples. Based on the results from Section 3.2, an adversarially trained model on the training set without any atypical samples has 95% clean accuracy and 55% adversarial accuracy on the test typical set. While, the model trained with 100% atypical samples only has 90.2%/50.4% clean/adversarial accuracy respectively. In other words, atypical samples act more like *"poisoning data"* (Biggio et al., 2012) to deteriorate model performance on typical samples. Furthermore, our study in Section 3.2 also demonstrates that this poisoning effect is absent in traditional ERM, where fitting atypical samples will not reduce the model accuracy. To deepen our understanding on this finding, we build a theoretical analysis based on Gaussian mixture models. We prove that, given certain atypical samples, any model which fits its adversarial counterpart (adv. training) must have a poor accuracy on typical samples. While, under the same setting, the models only fitting the clean version of this atypical sample (traditional ERM) can achieve optimal accuracy (i.e. $> 99\%$). Our empirical and theoretical results highlight the key difference between the effect of memorization in adversarial training and traditional ERM.

Motivated by our findings, we propose a novel algorithm called *Benign Adversarial Training (BAT)*, which can eliminate the negative influence from memorizing those "poisoning" atypical samples, meanwhile preserving the model's ability to memorize those "benign / useful" atypical samples. It is worth mentioning that, by fitting those "benign" atypical samples, the BAT method can achieve good clean accuracy on the atypical set; by eliminating those poisoning atypical samples, the BAT method can improve the clean & adv. accuracy on the typical set. Compared with PGD adversarial training (Madry et al., 2017) when it achieves the highest adversarial robustness, BAT has higher clean accuracy as well as adversarial accuracy. Compared to other popular variants of adversarial training such as (Zhang et al., 2019; Wang et al., 2019; Zhang et al., 2020), BAT is the only one obtaining both better (or comparable) clean and adversarial accuracy than (Madry et al., 2017), on complex datasets such as CIFAR100 and Tiny Imagenet (Le & Yang, 2015).

## 2 DEFINITION AND NOTATION

### 2.1 ATYPICAL SAMPLES AND MEMORIZATION

In this section, we start by introducing necessary concepts and definitions about the memorization effects. As well known, the practice of training deep neural networks (DNNS) cannot be well explained by standard theories about model generalization (Evgeniou et al., 2000; Bartlett & Mendelson, 2002). At a high level, the standard theories underline the importance of regularization on the model

complexity, to avoid "overfitting" the outliers and nonuseful samples. While, for DNNs, we always tune the model to hit almost perfect training accuracy and enjoy good test performance.

Fortunately, recent works (Feldman, 2020; Feldman & Zhang, 2020; Bartlett et al., 2020; Muthuku-mar et al., 2020) make significant progress to close this gap from both theoretical and empirical perspectives. They suggest that the memorization is one key property for DNNs to achieve optimal generalization performance. In detail, the works (Feldman, 2020; Feldman & Zhang, 2020) point out that the common benchmark datasets, such as CIFAR10, CIFAR100, ImageNet, contain a large portion of atypical samples (or namely, rare samples, sub-populations, etc.). These atypical samples look very different from the other samples in the main distribution of its labeled class, and are statistically indistinguished from outliers or mislabeled samples. DNNs can only fit these samples by memorizing their labels. Moreover, without memorizing these atypical samples during training, the DNNs can totally fail to predict the atypical samples in the test set (Feldman & Zhang, 2020).

**Identify Atypical Samples.** To identify such atypical samples in common datasets in practice, one representative strategy (Feldman & Zhang, 2020) proposes to examine which training samples can only be fitted by memorization, and measure each training sample's *"memorization value"*. Formally, for a training algorithm $\mathcal{A}$ (i.e., ERM), the memorization value "mem$(\mathcal{A}, \mathcal{D}, x_i)$" of a training sample $(x_i, y_i) \in \mathcal{D}$ in training set $\mathcal{D}$ is defined as:

$$\text{mem}(\mathcal{A}, \mathcal{D}, x_i) = \Pr_{F \leftarrow \mathcal{A}(\mathcal{D})} (F(x_i) = y_i) - \Pr_{F \leftarrow \mathcal{A}(\mathcal{D} \backslash x_i)} (F(x_i) = y_i), \tag{1}$$

which calculates the difference between the model $F$'s accuracy on $x_i$ with and without $x_i$ removed from the training set $\mathcal{D}$. In practice, (Feldman & Zhang, 2020) trains a DNN model using ERM method for 1,000 trials, with each trial preserves 70% samples of the whole training dataset. Based on this metric, one can identify all common datasets having a large fraction of atypical samples. For example, CIFAR10, CIFAR100 and Tiny ImageNet have more than 11%, 40% and 49% samples with a large memorization value $> 0.15$. The similar strategy can also facilitate to find atypical samples in the test set, which are the samples that are strongly influenced by atypical training samples. In detail, by removing an atypical training sample $(x_i, y_i)$, we calculate its *"influence value"* on each test sample $(x'_j, y'_j) \in \mathcal{D}'$ in test set $\mathcal{D}'$:

$$\text{infl}(\mathcal{A}, \mathcal{D}, x_i, x'_j) = \Pr_{F \leftarrow \mathcal{A}(\mathcal{D})} (F(x'_j) = y'_j) - \Pr_{F \leftarrow \mathcal{A}(\mathcal{D} \backslash x_i)} (F(x'_j) = y'_j). \tag{2}$$

If the sample pair $(x_i, x'_j)$ has a high influence value, removing the atypical sample $x_i$ will drastically decrease the model's accuracy on $x'_j$. In Appendix A, we provide more detailed discussions on the memorization effect and atypical samples, containing: (a) the distribution and frequency of atypical samples in common datasets and (b) alternative (more efficient) metrics to identify atypical samples, such as confidence-based method and ensemble disagreement (Carlini et al., 2019).

## 2.2 ADVERSARIAL TRAINING

Similar to classification models trained via empirical risk minimization (ERM) algorithms, adversarial training methods (Madry et al., 2017; Kurakin et al., 2016) are also devised to fit the whole dataset by training the model on manually generated adversarial examples. Formally, they are optimized to have the minimum adversarial loss:

$$\min_F \mathbb{E}_x \left[ \max_{||\delta|| \leq \epsilon} \mathcal{L}(F(x + \delta), y) \right], \tag{3}$$

which is the model $F$'s average loss on the data $(x, y)$ that perturbed by adversarial noise $\delta$. These adversarial training methods have been shown to be one of the most effective approaches to improve the model robustness against adversarial attacks. Note that similar to traditional ERM, adversarially trained models can also achieve very high training performance. For example, under WideResNet28-10 (WRN28), PGD adversarial training (He et al., 2016) can achieve 100% clean accuracy and 99% adversarial accuracy. It suggests these DNN models have sufficient capacity to memorize the labels of these atypical samples and their adversarial counterparts. However, different from ERM, adversarially trained models usually suffer from bad generalization performance during test. For example, the WRN28 model can only have 59% and 24% test clean accuracy and adversarial accuracy. Moreover, the study (Rice et al., 2020) suggests that during the adversarial training process, the model's test adversarial accuracy keeps dropping as more training data is fitted (after the first time learning rate

decay). Thus, these facts indicate that the memorization in adversarial training is probably not always beneficial to the test performance and requires deep understanding. In the following section, we will draw key connections between these properties with the memorization effect.

## 3  ATYPICAL SAMPLES IN ADVERSARIAL TRAINING

In this section, we attempt to understand adversarial training's behavior by studying its relation with the memorization effect. The discussions are based on PGD-adversarial training (Madry et al., 2017) on CIFAR 100. Implementation details and results on more datasets, i.e., CIFAR 10 and Tiny ImageNet (Le & Yang, 2015) are shown in Appendix B where we make similar observations.

### 3.1  ADVERSARIAL ROBUSTNESS OF ATYPICAL SAMPLES CAN HARDLY GENERALIZE

In this subsection, we first check whether fitting atypical samples in adversarial training can effectively help the model correctly and robustly predict the atypical samples in the test set. We apply PGD adversarial training (Madry et al., 2017) on original CIFAR100 dataset for 200 epochs and evaluate the model's clean accuracy and adversarial accuracy on training atypical set $\mathcal{D}_{\mathrm{atyp}} = \{x_i \in \mathcal{D} : \mathrm{mem}(x_i) > 0.15\}$ and its corresponding test atypical set $\mathcal{D}'_{\mathrm{atyp}} = \{x'_j \in$

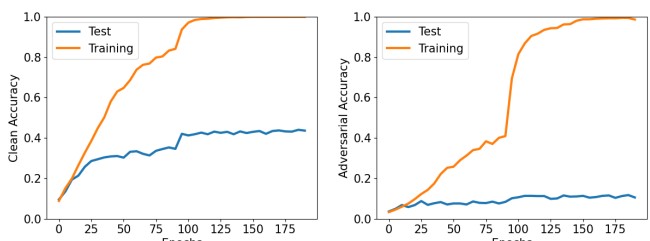

Figure 1: Clean Acc. (left) & Adv. Acc (right) on **Atypical** Set of CIFAR100 under WRN28

$\mathcal{D}' : \mathrm{infl}(x_i, x'_j) > 0.15, \text{for } \forall x_i \in \mathcal{D}_{\mathrm{atyp}}\}$. In Figure. 1, we report the algorithm's performance (clean & adv. acc.) on these atypical sets along with the training process. From the results, we observe that the WRN28 model is capable to memorize almost all atypical samples and their adversarial counterparts, since it can achieve $\approx 100\%$ clean & adversarial accuracy on the training atypical set. As the training goes, the models' clean accuracy on the test atypical set gradually improves and finally approaches 41%. However, its adversarial robustness keeps constant around 10% from the beginning epochs to the last ones, no matter how high the training performance is. These results suggest that memorizing atypical samples in adversarial training may only improve their test clean accuracy, but hardly help their adversarial robustness. Recall that in CIFAR100, atypical set $\mathcal{D}_{\mathrm{atyp}}$ (with memorization value $> 0.15$) covers 40% samples of the whole dataset. Completely failing on the adversarial robustness of atypical samples could be one key reason that contributes to the poor robustness generalization of DNNs (Rice et al., 2020).

As the previous theoretical study (Schmidt et al., 2018) stated, for a model to achieve good robustness generalization performance, it always demands a training set with much more samples, than a model to have good clean accuracy generalization. In our case, the sub-population of each particular atypical sample has a very low frequency to appear in the training set, and it is always deviated from the main sub-population. Thus, in the sub-population of this atypical sample, it is equivalent to a classification task based on an extremely small dataset, with one or a few training samples given. Thus, the robustness of atypical samples can be extremely hard to generalize.

### 3.2  MEMORIZING ATYPICAL SAMPLES HURTS TYPICAL SAMPLES' PERFORMANCE

In this subsection, we find that fitting atypical samples will even bring negative effects on "typical" samples. Here, we define "typical" samples as the subset of training set $\mathcal{D}$ which has low memorization values: $\mathcal{D}_{\mathrm{typ}} = \{x_i \in \mathcal{D} : \mathrm{mem}(x_i) < 0.02\}$. For the test typical set $\mathcal{D}'_{\mathrm{typ}}$, we exclude all test samples which have high influence values from any atypical training samples, and also exclude the samples that have low success rate to predict using ERM algorithm $\mathcal{A}$ (the samples which cannot be learned from $\mathcal{D}$): $\mathcal{D}'_{\mathrm{typ}} = \mathcal{D}' - \{x'_j : \mathrm{infl}(x_i, x'_j) > 0.02, \text{for } \forall x_i \in \mathcal{D}_{\mathrm{atyp}}\} \cup \{x'_j : \mathrm{Pr}_{F \leftarrow \mathcal{A}(\mathcal{D})}(F(x'_j) = y_j) < 0.8\}$.

We conduct PGD adversarial training (Madry et al., 2017) for several trials on resampled CIFAR100 datasets: each dataset is constructed with the whole training typical set $\mathcal{D}_{\mathrm{typ}}$, and a part of the training atypical set $\mathcal{D}_{\mathrm{atyp}}$ (randomly sample 0%, 20% and 100% in $\mathcal{D}_{\mathrm{atyp}}$). In Fig. 2, we report the adversarially trained model's clean and adversarial accuracy on the test typical set $\mathcal{D}'_{\mathrm{typ}}$ and check the

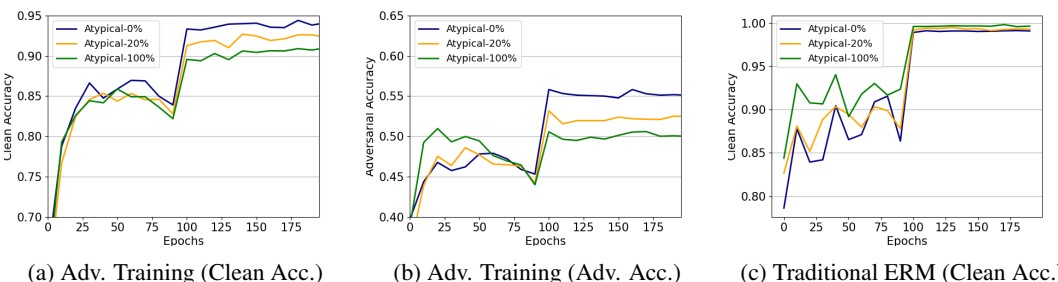

(a) Adv. Training (Clean Acc.)     (b) Adv. Training (Adv. Acc.)     (c) Traditional ERM (Clean Acc.)

Figure 2: Adversarial Training vs. Traditional ERM on **Typical** Set of CIFAR100 under WRN28

impact of atypical samples on the typical samples. From the results, we find that the existence of atypical samples makes a significant influence on the typical samples. For example, an adversarially trained model without atypical samples has 94.7% clean accuracy and 55.0% adversarial accuracy on the test typical samples (on the last epochs). While, the model trained with all atypical samples only has 90.2% and 50.4% clean & adv. accuracy, respectively. These results suggest: the more atypical samples exist in the training set, the poorer performance the model will have on $\mathcal{D}'_{\text{typ}}$. In other words, these atypical samples act like "poisoning" samples (Biggio et al., 2012; Xu et al., 2019) which can deteriorate the model's performance on typical samples. It is also worth mentioning that this poisoning phenomenon does not appear in the traditional ERM algorithm. From Figure 2c, we can see that the models have equal accuracy on the typical set, when different numbers of atypical samples appear in the training set.

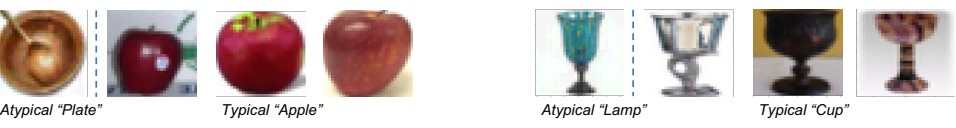

Atypical "Plate"     Typical "Apple"        Atypical "Lamp"     Typical "Cup"

Figure 3: Poisoning Atypical Samples

**Poisoning Atypical Samples** A natural question is what kinds of atypical samples are likely to "poison" model robustness and why? Different from previous literature about poisoning samples, which assume that poisoning samples are most mis-labeled samples (Li et al., 2020), CIFAR100 is a clean dataset with no or very few wrong labels. However, we hypothesize that the atypical samples which poison the model performance might pertain to some features of a "wrong" class. Recall that atypical samples are always distinct from the main data distribution in their labeled class, it is likely that they are closer to the distribution of a "wrong" class. As shown in Fig. 3, in CIFAR100 dataset, we identify an atypical "plate" which visually resembles images in "apple" (the examples are found via the method in Section 4.1). If the model memorizes this atypical "plate" and predicts any samples with similar features to be "plate", the model cannot distinguish between "apple" and "plate".

Why does this poisoning effect only happen during adversarial training, instead of traditional ERM? To deepen our understanding of this question, we theoretically analyze the consequence of fitting "poisoning" atypical samples for both algorithms. In particular, we formally state: *the poisoning effect during adversarial training can be significantly stronger than traditional ERM*. Our analysis is based on a binary classification task with a two-dimensional Gaussian Mixture distribution:

$$y \overset{u.a.r}{\sim} \{-1, +1\}, \quad \theta = [\eta, 0], \eta \in \mathbb{R}_+, \quad x \sim \begin{cases} \mathcal{N}(\theta, \sigma^2 I) & \text{if } y = +1 \\ \mathcal{N}(-\theta, \sigma^2 I) & \text{if } y = -1, \end{cases} \quad (4)$$

and we consider linear models $F = \text{sign}(w^T x + b)$, where $w$ and $b$ are the model weight and bias. Moreover, we assume there is an atypical training sample $x_0 \in \mathbb{R}^2$ labeled as "$-1$", with Euclidean distance to the center of each class $||x_0 - (-\theta)||_2 = d_{-1}, ||x_0 - \theta||_2 = d_{+1}$. Intuitively, this setting of $x_0$ resembles the cases in Figure 3 if $d_{+1}$ is relatively small, since it has a short distance to the wrong class. We also constrain $d_{-1} \leq 2 \times \eta$ for $x_0$ to preserve certain similarity to the class "-1". The following theorem shows that there always exists a model which fits the sample $x_0$, and achieve optimal accuracy (i.e. > 99%) on both classes, although $d_{+1}$ can be small.

**Theorem 1 (Fitting Atypical Samples in traditional ERM)** *For a data distribution in Eq. 4, given that $\eta/\sigma$ is large enough, we define $l = \sqrt{2} \cdot \Phi^{-1}(0.99) \cdot \sigma \cdot (1 + \Phi^{-1}(0.99) \cdot \frac{\sigma}{\eta})^{-1/2}$. For a point $x_0$ with $d_{-1} \leq 2\eta$, if $d_{+1} \geq l$, then there exist a model $F$ fitting $x_0$: $F(x_0) = -1$ and has classification error at most $1\%$ on both classes.*

The detailed proof can be found in Appendix C. Remarkably, for classification tasks if the two classes are well separated, i.e. $\sigma/\eta$ is sufficiently small, there exist models that can achieve high performance even when the atypical sample $x_0$ is only around $(\sqrt{2} \cdot \Phi^{-1}(0.99) \cdot \sigma)$ distant away from the center of class "+1", which is much smaller than the center distance of the two class $2\eta$. This suggests that for traditional ERM, the poisoning effect of fitting the atypical sample $x_0$ is negligible, even if $x_0$ is close to the wrong class "+1". Next, we show that under the same setting, any model which fits the adversarial counterpart of $x_0$, must have a poor accuracy on the class "+1".

**Theorem 2 (Fitting Atypical Samples in Adversarial Training)** *For the same data distribution in Eq. 4 and same definition $l = \sqrt{2} \cdot \Phi^{-1}(0.99) \cdot \sigma \cdot (1 + \Phi^{-1}(0.99) \cdot \frac{\sigma}{\eta})^{-1/2}$. We assume $l \leq d_{+1} < k \cdot l$, where $1 \leq k << \eta/\sigma$. Then, any model $F$ will have a classification error at least $50\%$ on class "+1", when $F$ fits the adversarial example of $x_0$: $F(x_0 + \delta) = -1, \forall ||\delta||_2 \leq \epsilon$, and*

$$\epsilon \geq \sigma \cdot (k \cdot \sqrt{2} \cdot \Phi^{-1}(0.99)) \cdot (1 + \Phi^{-1}(0.99) \cdot \frac{\sigma}{\eta})^{-1/2} \tag{5}$$

Notably, when the two classes are well separated, i.e. $\sigma/\eta$ is small, the adversarial training bound $\epsilon$ is also much smaller than $\eta$. The analysis suggests that when $x_0$ is close to the class "+1", all linear models which fit the adversarial example of $x_0$ will have a large error on class "+1", even the adversarial training bound $\epsilon$ is small. Thus, in adversarial training, the poisoning effect of fitting atypical samples is significantly greater than that in traditional ERM. These theorems highlight the key difference between adversarial training and traditional ERM when fitting atypical samples, which motivate us to devise strategies to eliminate this poisoning effect during adversarial training.

## 4 BENIGN ADVERSARIAL TRAINING

Based on previous discussions, the effect of atypical samples in adversarial training can be briefly summarized as: **(a)** They can benefit the models' clean accuracy (especially on atypical samples), but hardly improve their robustness. **(b)** They hurt the performance of typical samples. In this section, we ask the question: *Can we eliminate the negative influence from fitting atypical samples during adversarial training?* Admittedly, strategies for adversarial training to stop at early epoch (Rice et al., 2020) is potential to achieve this goal, since the atypical samples may be fitted in the later stages of training. However, it is still unclear how to properly identify such an epoch in general scenarios. Moreover, the early-stop mechanism tends to ignore all atypical samples and significantly limits the model's ability to handle atypical samples, especially on complex datasets such as CIFAR100 and ImageNet with large fractions of atypical samples.

In this work, we propose a novel algorithm *Benign Adversarial Training (BAT)*. It is composed of two major components: **(i)** *Reweighting* to mitigate the impact from poisoning atypical samples and learn benign atypical samples; **(ii)** *Discrimination Loss* to further protect the discrimination ability among different classes. Combining these two components, BAT can improve the performance of typical samples, but still preserve the ability to fit those "useful"' atypical samples. Compared to PGD adversarial training (Madry et al., 2017), BAT enjoys better clean vs. adversarial accuracy trade-off. More results can be found in Section 5. Next, we detail these two components of BAT.

### 4.1 REWEIGHTING & POISONING SCORE

Following PGD adversarial training (Madry et al., 2017), BAT starts by fitting manually generated adversarial examples using Projected Gradient Descent (PGD). During the training process, in order to identify which samples can poison or degrade the model's performance, we define the "*poisoning score*" for each training (adversarial) sample $x_i^{\text{adv}}$ as:

$$\mathbf{q}(x_i^{\text{adv}}) = \max_{t \neq y} F_t(x_i^{\text{adv}}) \tag{6}$$

which is the model $F$'s largest prediction score (after softmax) to a wrong class $t$ other than the true class $y$. A high poisoning score suggests that the current model predicts the sample $x^{\text{adv}}$ to be a wrong

class with high confidence. Note that the atypical samples are usually fitted into the model later than typical samples. Under a model with a few atypical samples fitted, if there is an atypical sample $x_i^{\text{adv}}$ with high $\mathbf{q}(x_i^{\text{adv}})$, $x_i^{\text{adv}}$ is very likely to be close to the distribution of a wrong class instead of its true class. In Figure. 3, we present several atypical (adversarial) samples with poisoning scores larger than 0.8. These samples present semantic features which are very similar to the wrong class that the model predicts. Therefore, fitting these atypical samples could cause the model to learn mixed concepts of features and degrades the model performance.

During the training process, we desire that the model should mitigate the influence from the atypical samples with large poisoning scores, but still learn other "useful /benign" atypical samples. Thus, we design a cost-sensitive reweighting strategy which downweights the cost of atypical samples with large poisoning scores. In particular, we specify the weight value $w_i$ for each training sample $x_i^{\text{adv}}$ as:

$$w_i = \begin{cases} \exp(-\alpha \cdot \mathbf{q}(x_i^{\text{adv}})) & \text{if } \operatorname{mem}(x_i) > \sigma \text{ and } \arg\max_t F_t(x_i^{\text{adv}}) \neq y \\ 1 & \text{Otherwise.} \end{cases} \tag{7}$$

where $\alpha \in \mathbb{R}^+$ and $\sigma$ control the size of the reweighted atypical set. Since the function $\exp(-\alpha \cdot (\cdot))$ is decreasing and ranges from 1 to 0, the samples with large poisoning scores will be assigned with small weights close to 0. Thus, in the reweighting algorithm, we train the model to find optimal model $F_{\text{rw}}$ by assigning the weight vector $w$:

$$F_{\text{rw}} = \arg\min_F \frac{1}{\sum_i w_i} \sum_i \left[ w_i \cdot \mathcal{L}(F(x_i^{\text{adv}}), y_i) \right] \tag{8}$$

In the training process, those (adversarial) training samples with large poisoning scores are assigned with small weights and correspondingly their influence will be largely mitigated. Note that in Eq. 7, estimating $\operatorname{mem}(x_i)$ for each $x_0$ via the leave-out retraining method described in Eq. 1 is potential to be computational costly. However, alternative efficient strategies have been discussed to estimate how likely one sample is atypical, including confidence-based strategies and ensemble disagreement (Carlini et al., 2019). These methods are shown to have good alignment with the method in Eq. 1, thus these strategies can be leveraged to improve the efficiency of Eq. 7. In Appendix A, we provide detailed discussions about these methods.

### 4.2 DISCRIMINATION LOSS.

In addition, we introduce *Discrimination Loss* to further eliminate the poisoning effect of fitting atypical samples. Based on the theoretical analysis Section 3.2, imagine that the atypical sample $x_0$ has a fixed distance $d_{-1}$ to the center of its labeled class, one strategy to mitigate the poisoning effect of fitting $x_0$ is to increase the two class's center distance $2\eta$ and hence increase the distance $d_{+1}$ of the poisoning example to the poisoned class "+1". Motivated by this point, we introduce $\mathcal{L}_{DL}$ to increase the distances of typical samples among different classes in the representation space.

$$\mathcal{L}_{DL}(F) = \mathop{\mathbb{E}}_{\substack{(x_i, y_i), (x_j, y_j) \\ \{(x_k, y_k)\}_{k=1}^K}} \left[ -\log \frac{e^{h(x_i^{\text{adv}}) \cdot h(x_j^{\text{adv}})/\tau}}{\sum_{k=1}^K e^{h(x_i^{\text{adv}}) \cdot h(x_k^{\text{adv}})/\tau}} \right]. \tag{9}$$

$$\text{where } y_i = y_j; \quad y_k \neq y_i, \forall k = 1, 2, ...K; \operatorname{mem}(x_i), \operatorname{mem}(x_j), \operatorname{mem}(x_k) < \sigma$$

Specifically, $h(\cdot)$ is the model $F$'s pen-ultimate layer's output representation; $\tau \in \mathbb{R}^+$ is the temperature value and $K \in \mathbb{Z}^+$ is a fixed number of the samples that are randomly chosen with different labels with $y_i$[2]. Intuitively, constraining the *Discrimination Loss* imposes the model to output similar representations for the typical sample pairs $(x_i^{\text{adv}}, x_j^{\text{adv}})$ in the same class, and distinct representations for $(x_i^{\text{adv}}, x_k^{\text{adv}})$ from different classes. Similar approaches have been widely used in representation learning, such as Triplet Loss (Schroff et al., 2015) and contrastive learning methods (Chen et al., 2020), which stress the good property of DNNs' representation space. With the Discrimination Loss incorporated, the final objective of BAT is formulated as:

$$F_{\text{bat}} = \arg\min_F \left( \frac{1}{\sum_i w_i} \sum_i \left[ w_i \cdot \mathcal{L}(F(x_i^{\text{adv}}), y_i) \right] + \beta \cdot \mathcal{L}_{DL}(F) \right) \tag{10}$$

where $\beta > 0$ that controls the intensity of Discrimination Loss. The detailed training scheme of Benign Adversarial Training is presented in Appendix D.

---

[2]In practice, under training algorithms using mini-batches, the set $\{x_k\}_{k=1}^K$ can be replaced by all (typical) samples in the mini-batch, with different labels from $y_i$.

## 5 EXPERIMENT

In this section, we present the experimental results to validate the effectiveness of the proposed BAT algorithm on benchmark datasets and compare it with state-of-the-art baseline methods. The implementation can be found via `https://anonymous.4open.science/r/benign-adv-77C5`.

### 5.1 EXPERIMENTAL SETUP

In this work, in order to demonstrate the merit of BAT, we conduct the experiments mainly on benchmark datasets CIFAR100 (Krizhevsky et al., 2009) and Tiny ImageNet (Le & Yang, 2015), which are relatively complex datasets (i.e., containing larger fractions of atypical samples). For both datasets, we study the algorithms under the model architectures ResNet18 and WideResNet (WRN) (He et al., 2016). In this section, we only present the results of WRN models and leave the results on ResNet18 in Appendix E. As a fair comparison with BAT, we implement the baseline algorithms including PGD adversarial training (Madry et al., 2017) as well as its most popular variant TRADES (Zhang et al., 2019). In addition, we include several recent algorithms: MART (Wang et al., 2019) and GAIRAT (Zhang et al., 2020), which also incorporate reweighting strategies into adversarial training. For BAT and all baseline methods, we run the algorithms using SGD (Bottou, 2010) for 160 epochs where the learning rate is from 0.1 and decays by 0.1 after the epoch 80 and 120. More implementation details are listed in Appendix E.

**Performance on CIFAR100.** For a comprehensive comparison between different methods on CIFAR100, in Table 1, we report the models' clean accuracy and adversarial accuracy against $l_\infty$-8/255 adversarial examples, generated by various attacking algorithms including FGSM attack (Goodfellow et al., 2014), PGD attack (Madry et al., 2017), CW (Carlini & Wagner, 2017) and Auto-Attack (Croce & Hein, 2020)). For BAT, we report its performance when choosing its optimal hyperparameter: $\alpha = 1$, $\beta = 0.2$ and $\alpha = 2$, $\beta = 0.2$. In Section 5.2, we will discuss the impact of the selection of $\alpha$ and $\beta$ on BAT and argue that both components are necessary for BAT to achieve the optimal performance. For baseline methods, the settings follow the original papers' suggestions and their results are reported on the checkpoints with the best PGD adversarial accuracy.

Table 1: Performance of BAT vs. Baselines on CIFAR100 Under WRN28

| Method | Clean Acc. | FGSM | PGD | CW | AA. |
|---|---|---|---|---|---|
| PGD Adv. Training | 59.7 (-2.3) | 34.9 (-3.7) | 24.7 (-3.8) | 24.2 (-2.3) | 22.5 (-2.3) |
| TRADES ($1/\lambda = 5$) | 57.3 (-4.7) | 34.5 (-4.1) | 24.9 (-3.6) | 24.6 (-1.9) | 22.9 (-1.9) |
| MART (Wang et al., 2019) | 56.5 (-5.5) | 36.1 (-2.5) | 26.8 (-1.7) | 25.3 (-1.2) | 23.8 (-1.0) |
| GAIRAT (Zhang et al., 2020) | 60.2 (-1.8) | 34.8 (-3.8) | 24.4 (-4.1) | 24.8 (-1.5) | 22.9 (-1.9) |
| BAT ($\alpha = 1, \beta = 0.2$) | **62.0** (+0.0) | 38.6 (+0.0) | **28.5** (+0.0) | **26.5** (+0.0) | **24.8** (+0.0) |
| BAT ($\alpha = 2, \beta = 0.2$) | 61.4 (-0.6) | **38.9** (+0.3) | 28.2 (-0.3) | 26.3 (-0.2) | **24.8** (-0.0) |

In Table 1, we also report the performance difference between baseline methods vs. BAT ($\alpha = 1$, $\beta = 0.2$) in parentheses. From the results, we can find that BATs enjoy good clean & adversarial accuracy trade-off at the same time. For example, BAT methods can achieve the highest clean accuracy as well as adversarial accuracy among all baseline methods. Compared to PGD adversarial training Madry et al. (2017), BAT methods have around 2 to 3% improvement on both clean accuracy and adversarial accuracy on strong attacks such as PGD, CW and AA. Compared to other baseline methods such as MART Wang et al. (2019), it has more than 1% robustness improvement evaluated by strong adversarial attacks, but much higher clean accuracy ($> 5\%$).

**Performance on Tiny ImageNet.** Tiny ImageNet (Le & Yang, 2015) contains 200 classes of the images in the original ImageNet (Krizhevsky et al., 2012) dataset, with 500 training images for each class, and image size $64 \times 64$. In our experiments, we only choose the first 50 classes in Tiny ImageNet for training and prediction. Since the image size is $64 \times 64$, for both training and robustness evaluation, we consider the adversarial attacks are bounded by $l_\infty$-norm-4/255. In Table 2, we report the performance of BAT and baseline methods. Similar to the conclusions we can make from CIFAR100, BATs can achieve consistently better clean accuracy as well as adversarial accuracy compared to baseline methods.

### 5.2 ABLATION STUDY

In this subsection, we study the potential impact of the hyperparameters chosen in BAT, which is $\alpha$ that controls the *Reweighting* process and $\beta$ that controls the *Discrimination Loss*. In Fig. 4, we

Table 2: Performance of BAT vs. Baselines on Tiny ImageNet Under WRN28

| Method | Clean Acc. | FGSM | PGD | CW | AA. |
|---|---|---|---|---|---|
| PGD Adv. Training | 58.9 (-3.4) | 35.7 (-5.9) | 31.7 (-4.1) | 30.0 (-3.7) | 30.0 (-4.3) |
| TRADES ($1/\lambda = 5$) | 59.7 (-2.6) | 37.4 (-4.2) | 32.0 (-3.8) | 31.8 (-1.9) | 32.0 (-2.3) |
| MART (Wang et al., 2019) | 58.2 (-4.1) | 41.0 (-0.6) | 35.6 (-0.2) | 33.9 (+0.2) | 34.1 (-0.2) |
| GAIRAT (Zhang et al., 2020) | 60.5 (-1.8) | 39.1 (-2.5) | 33.1 (-2.7) | 31.9 (-1.8) | 32.3 (-2.0) |
| BAT ($\alpha = 1, \beta = 0.2$) | 62.3 (+0.0) | 41.6 (+0.0) | 35.8 (+0.0) | 33.7 (+0.0) | 34.3 (+0.0) |
| BAT ($\alpha = 2, \beta = 0.2$) | **62.4** (+0.1) | **43.1** (+1.5) | **37.4** (+1.6) | **35.3** (+1.6) | **35.6** (+0.7) |

conduct the experiments on CIFAR100 with BAT when $\alpha$ is chosen in [0,1,2,3] and $\beta$ is in [0, 0.1, 0.2, 0.3]. In Fig. 4, we show each model's overall clean accuracy (left) & adversarial accuracy on PGD attack (right) along the Z-axis, and X-axis / Y-axis indicate the models' corresponding variables of $(\alpha, \beta)$. Note that when $\alpha = \beta = 0$, the BAT method regresses to original PGD adversarial training. From the result, we can see that a positive pair of $(\alpha, \beta)$ can benefit both model clean and adversarial accuracy (on PGD attack). Therefore, both components (i.e., *Reweighting* and $\mathcal{L}_{DL}$) of BAT are helpful and necessary. However, when $\alpha$ or $\beta$ is too large, it will hurt the BAT's clean accuracy. Hence, in CIFAR100, when $\alpha = 1$ or 2 and $\beta = 0.2$, BAT can achieve the optimal performance.

## 6 RELATED WORKS

**Memorization and atypical samples.** The memorization effect of overparameterized DNNs have been extensively studied both empirically (Zhang et al., 2016; Nakkiran et al., 2019) and theoretically (Bartlett & Mendelson, 2002). From traditional views, the memorization can be harmful to the model generalization, because it makes DNN models easily fit those outliers and noisy labels. How-

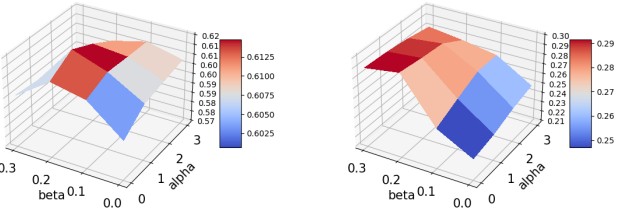

Figure 4: Clean Acc.(left) & Adv. Acc.(right) of BAT

ever, recent studies point out the concept of "benign overfitting" (Bartlett et al., 2020; Feldman, 2020; Feldman & Zhang, 2020), which suggests the memorization effect necessary for DNNs to have extraordinary performance on modern machine learning tasks. Especially, the recent work (Feldman & Zhang, 2020) empirically figures out those atypical/rare samples in benchmark datasets and show the contribution from memorizing atypical samples to the DNN's performance. Besides the work (Feldman & Zhang, 2020), there are also other strategies (Carlini et al., 2019) to find atypical samples in training dataset. Notably, our work is not the first effort to study the influence of memorization on DNN's adversarial robustness. A previous study (Sanyal et al., 2020) illustrates that memorizing the mis-labeled samples might be a reason to cause the DNNs' adversarial vulnerability. In our paper, we focus on atypical samples, which appear much more frequently in common datasets, and we study their impacts especially on adversarial training algorithms (Madry et al., 2017; Zhang et al., 2019; Chatterji & Long, 2020; Muthukumar et al., 2020).

**Adversarial robustness.** Adversarial training methods (Madry et al., 2017; Zhang et al., 2019; Wang et al., 2019; Zhang et al., 2016; Rice et al., 2020) are considered as one of the most reliable and effective methods to protect DNN models against adversarial attacks (Goodfellow et al., 2014; Xu et al., 2019). However, there are several intrinsic properties of adversarial training different from traditional ERM which require deeper understandings. For example, they always suffer from poor robustness generalization (Schmidt et al., 2018; Rice et al., 2020); they always present strong trade-off relation between clean accuracy vs. robustness (Tsipras et al., 2018; Zhang et al., 2019); they are likely to have biased accuracy & robustness among different subgroups of data (Xu et al., 2021). Our work aims to study the robustness generalization issues from the data perspective and demonstrate the significant connection of the memorization effect with these properties.

## 7 CONCLUSION

In this paper, we draw significant connections of the memorization effect of deep neural networks with the behaviors of adversarial training algorithms. Based on the findings, we devise a novel algorithm BAT to enhance the performance of adversarial training. The findings of the paper can motivate the futures studies in building robust DNNs with more attention on the data perspective.

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

# Appendix

In the supplementary materials, we provide more details about atypical samples and the proposed algorithm, as well as the full experimental results of the preliminary study and the proposed method.

## A    ADDITIONAL INTRODUCTION OF ATYPICAL SAMPLES

In this section, we provide additional introductions about the memorization effect and atypical samples in image classification tasks. We first introduce several strategies to identify atypical samples in common datasets. Then, we show several examples to illustrate the distribution & frequency in common datasets.

### A.1    HOW TO ESTIMATE WHETHER ONE SAMPLE IS ATYPICAL?

**Leave-out Method** The work (Feldman & Zhang, 2020) proposes to examine which training samples can only be fitted by memorization, and measure each training sample's *"memorization value"*. Formally, for a training algorithm $\mathcal{A}$ (i.e., ERM), the memorization value "mem$(\mathcal{A}, \mathcal{D}, x_i)$" of a training sample $(x_i, y_i) \in \mathcal{D}$ in training set $\mathcal{D}$ is defined as:

$$\text{mem}(\mathcal{A}, \mathcal{D}, x_i) = \Pr_{F \leftarrow \mathcal{A}(\mathcal{D})} (F(x_i) = y_i) - \Pr_{F \leftarrow \mathcal{A}(\mathcal{D} \setminus x_i)} (F(x_i) = y_i),$$

which calculates the difference between the model $F$'s accuracy on $x_i$ with and without $x_i$ removed from the training set $\mathcal{D}$ of algorithm $\mathcal{A}$. Note that for each sample $x_i$, if its memorization value is high, it means that removing $x_i$ from training data will cause the model with a high possibility to wrongly classify itself, so $x_i$ is very likely to be fitted only by memorization and be atypical. Although promising, the computational efficiency of Leave-out method is potential to be low. For example, the Leave-Out method requires repeatedly training a DNN model for 1,000 trials to accurately estimate the memorization values.

**Ensemble Disagreement** The work (Carlini et al., 2019) provides an alternative way to efficiently figure out the examples which are poorly represented by other samples from the training dataset (atypical samples). They assume the samples which are well-represented in the distribution should be relatively easy for many types of DNN models to learn. Concretely, the ensemble agreement trains a series of models $\mathcal{F} = \{F_j\}_{j=1}^m$ with different architectures and initializations. Then, for each sample $x_i$, it computes the disagreement of all models:

$$\text{ed}(x_i) = \frac{1}{m^2} \sum_{j=1}^m \sum_{p=1}^m \text{JS-Divergence}(F_j(x_i), F_p(x_i))$$

Intuitively, if ed$(x_i)$ is large, different models will give different predictions to the sample $x_i$. It suggest that the sample $x_i$ cannot be easily learned by most of models in $\mathcal{F}$. Therefore, the sample $x_i$ is likely to be atypical. In practice, (Carlini & Wagner, 2017) trains multiple DNN models for 10 random initializations, or get 10 pretrained from public resources, which significantly improve the efficiency to find atypical samples.

**Confidence-Based Method** Similar to the ensemble disagreement method, the work Carlini et al. (2019) assumes that the models in $\mathcal{F} = \{F_j\}_{j=1}^m$ should be more confident on examples that are

well-represented by the whole training dataset. Thus, the propose to measure the average confidence value of these models on sample $x_i$:

$$\text{conf}(x_i) = \frac{1}{m} \sum_{j=1}^{m} \max(F_j(x_i)[y_i]))$$

The smaller the confidence value of a sample $x_i$, it is more likely to be atypical. Remarkably, the work Carlini et al. (2019) empirically demonstrates that the three metrics decribed above have good alignment for datasets such as MNIST, CIFAR10, CIFAR100 (Krizhevsky et al., 2009) and ImageNet (Krizhevsky et al., 2012). Therefore, for practical application of our proposed method BAT, we can also substitute the leave-out method to ensemble disagreement and confidence-based methods, in order to to enhance the efficiency of BAT.

## A.2 Distribution of Atypical Samples

In Fig. 5, we provide several examples of images from CIFAR10, CIFAR100 (Krizhevsky et al., 2009) and Tiny ImageNet (Le & Yang, 2015) respectively, with different memorization value (as defined in Section 2.1) around $0.0, 0.5, 1.0$. These examples suggest that if the memorization value of an image is large, this image is very likely to be "atypical", as it presents very distinct semantic features with the images in the main distribution (with memorization value 0.0).

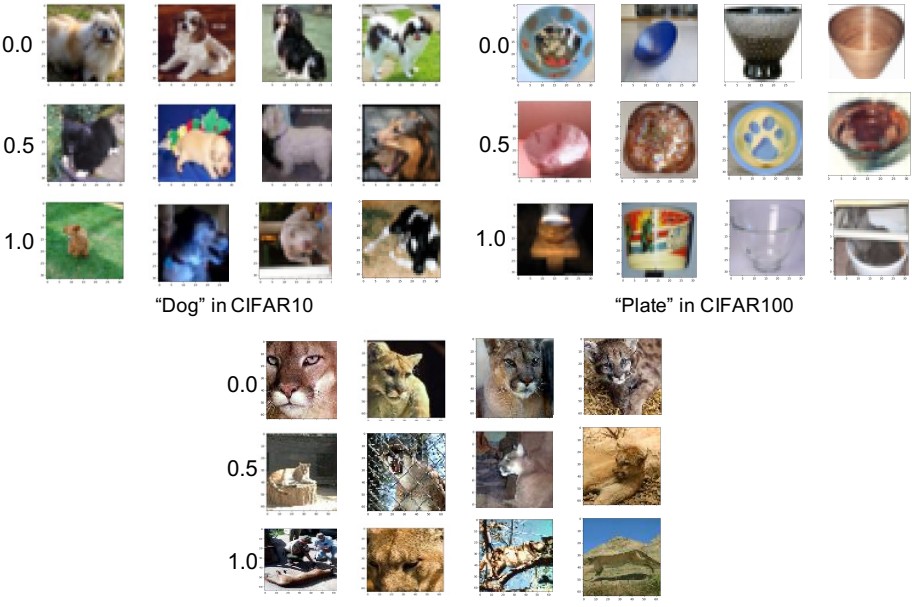

Figure 5: Examples of Images with Different Memorization Values

In Fig. 6, we provide histograms to show the distribution of the estimated memorization values of all training samples from CIFAR10, CIFAR100 and Tiny ImageNet. From Fig. 6, we can observe that atypical samples (with high memorization value > 0.15) consist of a significant fraction (over 40% & 50% respectively) in CIFAR100 and Tiny ImageNet. In CIFAR10, they also consist of a non-ignorable fraction which is over 10%.

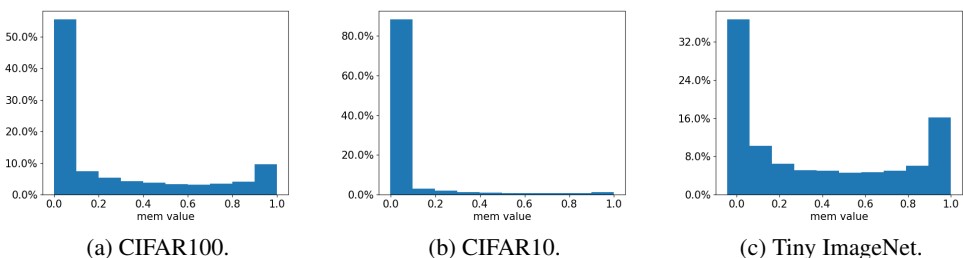

(a) CIFAR100.     (b) CIFAR10.     (c) Tiny ImageNet.

Figure 6: Frequencies of Training samples with Different **Memorization Values** in Various Datasets

In Fig. 7, we provide several pairs of images with high influence value (as defined in Section 2.1) which is over 0.15. In each pair, the training sample also has a high memorization value over 0.15. These examples suggest that there exist atypical samples in both training & test sets of CIFAR10, CIFAR100 and Tiny ImageNet. A pair of atypical samples (in the training set and test set) with a high influence value are visually very similar. Moreover, since they have high influence values, removing the atypical samples in the training set is very likely to cause the model to fail on the test atypical samples. Therefore, without memorizing the atypical sample in the training set, the model can hardly predict the atypical samples in the test set.

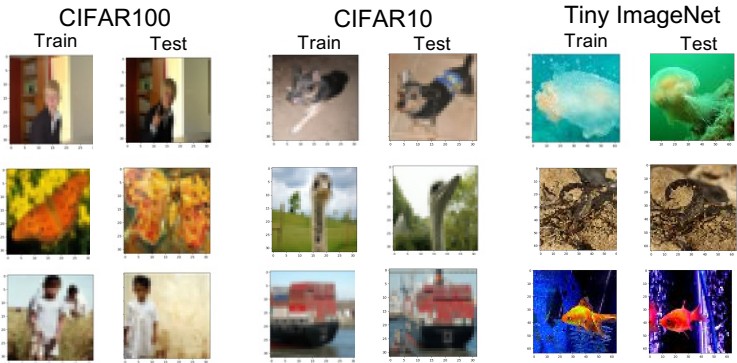

Figure 7: High Influence Pairs with Influence Value > 0.15

# B ADDITIONAL RESULTS FOR PRELIMINARY STUDY

In this section, we provide the full results of the preliminary study in Section 3 on CIFAR10, CIFAR100 and Tiny ImageNet, to illustrate the distinct behaviors of the memorization effect between traditional ERMs and adversarial training. In both ERM and adversarial training, we train the models under ResNet18 and WideResNet28-10 (WRN28) architectures. In the experiments, we train the models for 200 epochs with learning rate 0.1, momentum 0.9, weight decay 5e-4, and decay the learning rate by 0.1 at the epoch 150 and 200. For adversarial training, we conduct experiments using PGD adversarial training Madry et al. (2017) by default to defense against $l_\infty$-8/255 adversarial attack, with the exception on Tiny ImageNet, which is against $l_\infty$-4/255 attack. For robustness evaluation, we conduct a 20-step PGD attack.

### B.1 ADDITIONAL RESULTS FOR PRELIMINARY STUDY - SECTION 3.1

In this subsection, we provide more experimental results to validate the statement in Section 3.1, where we state that fitting atypical samples in adversarial training can only improve the clean accuracy of test atypical samples, but hardly help their adversarial robustness. We provide full empirical results to show that: **(i)** In traditional ERM, fitting atypical samples improves the clean accuracy of test atypical samples. **(ii)** In adversarial training, fitting (adversarial) atypical samples improves the clean accuracy of test atypical samples but can hardly improve the adversarial robustness of them. The experimental setting follows Section 3.1, where we apply traditional ERM and adversarial training on original CIFAR10, CIFAR100, Tiny ImageNet datasets. We evaluate the model's clean accuracy and adversarial accuracy on training atypical set $\mathcal{D}_{\text{atyp}} = \{x_i \in \mathcal{D} : \text{mem}(x_i) > 0.15\}$ and its corresponding test atypical set $\mathcal{D}'_{\text{atyp}} = \{x'_j \in \mathcal{D}' : \text{infl}(x_i, x'_j) > 0.15, \text{for } \forall x_i \in \mathcal{D}_{\text{atyp}}\}$.

**(i) Additional Results for Preliminary Study - Section 3.1 In Traditional ERM**

Fig. 8, Fig. 9 and Fig. 10 report the performance (clean accuracy) of traditional ERM, which is evaluated on atypical sets under ResNet18 (left) and WRN28 (right) on CIFAR100, CIFAR10 and Tiny ImageNet. From the figures, we can obverse that fitting atypical samples during training can effectively help the models to achieve good clean accuracy on test atypical samples in all datasets. Note that here we only report clean accuracy as they are not robust against adversarial attacks.

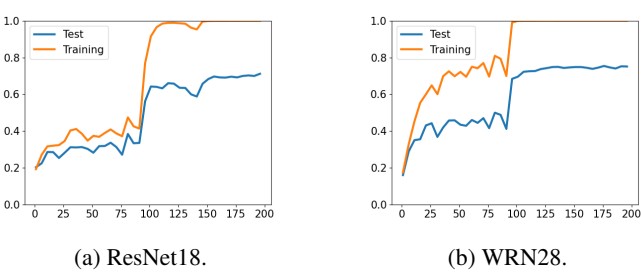

(a) ResNet18.      (b) WRN28.

Figure 8: Clean Accuracy on **Atypical** Set of CIFAR100

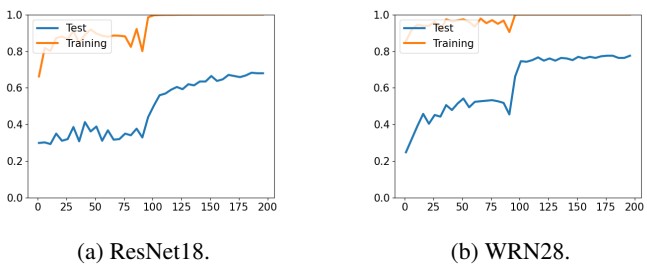

(a) ResNet18.      (b) WRN28.

Figure 9: Clean Accuracy on **Atypical** Set of CIFAR10

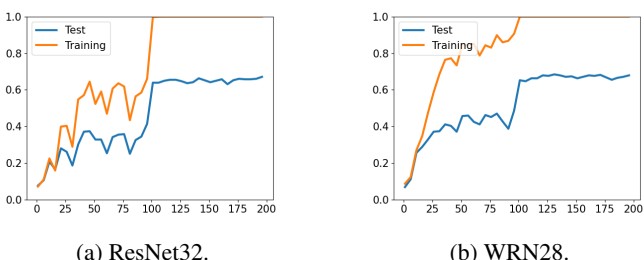

(a) ResNet32.      (b) WRN28.

Figure 10: Clean Accuracy on **Atypical** Set of Tiny ImageNet

**(ii) Additional Results for Preliminary Study - Section 3.1 In Adversarial Training**

Fig. 11, Fig. 12 and Fig. 13 report the performance of adversarially trained models. We evaluate the clean accuracy and adversarial accuracy on the training atypical set $\mathcal{D}_{\text{atyp}}$ and test atypical set $\mathcal{D}'_{\text{atyp}}$. From the results, we can observe that although fitting atypical samples can help the model to have modest clean accuracy on test atypical samples, the adversarial robustness of them is constantly low and can hardly be improved during the whole training process.

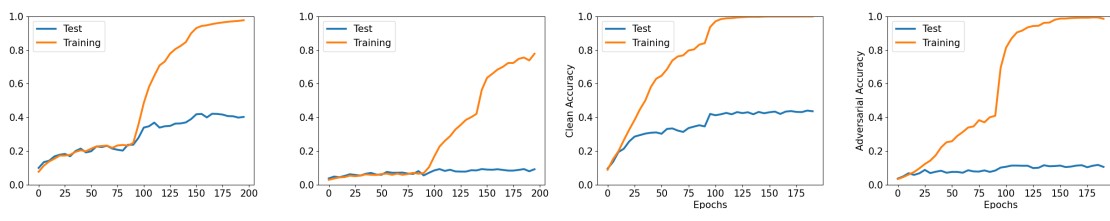

(a) Clean (left) & Adv Acc. (right) under ResNet18.    (b) Clean (left) & Adv Acc. (right) under WRN28.

Figure 11: Clean Accuracy and Adversarial Accuracy on **Atypical** Set of CIFAR100

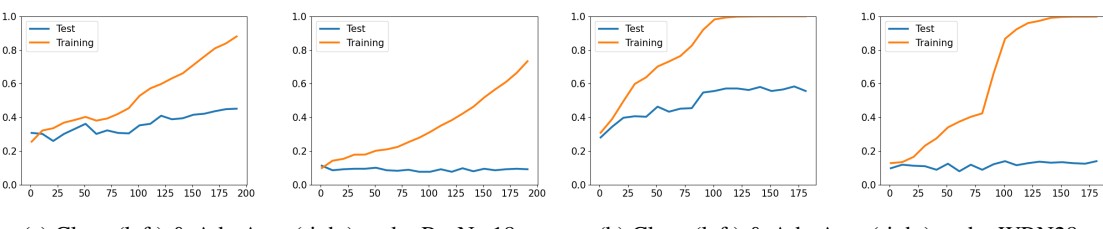

(a) Clean (left) & Adv Acc. (right) under ResNet18.    (b) Clean (left) & Adv Acc. (right) under WRN28.

Figure 12: Clean Accuracy and Adversarial Accuracy on **Atypical** Set of CIFAR10

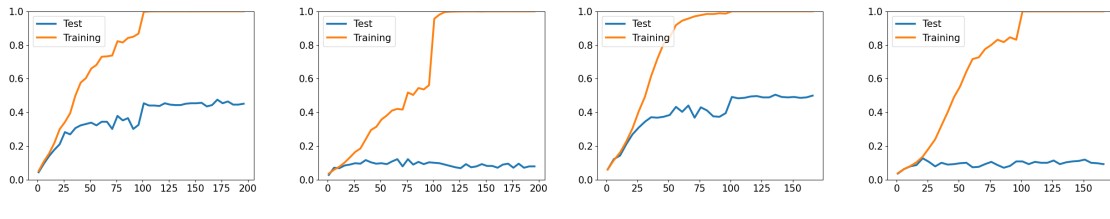

(a) Clean (left) & Adv Acc. (right) under ResNet32.    (b) Clean (left) & Adv Acc. (right) under WRN28.

Figure 13: Clean Accuracy and Adversarial Accuracy on **Atypical** Set of Tiny ImageNet

### B.2 ADDITIONAL RESULTS FOR PRELIMINARY STUDY - SECTION 3.2

In this subsection, we provide more experimental results to validate the statement in Section 3.2, where we state that fitting atypical samples in adversarial training can hurt the performance (clean & adversarial accuracy) of typical samples. We provide full empirical results to show that: **(i)** In traditional ERM, fitting atypical samples will not hurt the models' performance (clean accuracy) of typical samples. **(ii)** In adversarial training, fitting atypical samples can degrade the clean & adversarial accuracy of typical samples. **(iii)** In adversarial training, fitting atypical samples can degrade the quality of learned representations, especially the models' discrimination between classes. The experimental setting follows Section 3.2, where we train the models for several trails on resampled (CIFAR100, CIFAR10, Tiny ImageNet) datasets: each dataset is constructed with the whole training typical set $\mathcal{D}_{\text{typ}}$, and a part of the training atypical set $\mathcal{D}_{\text{atyp}}$ (randomly sample 0%, 20% and 100% in $\mathcal{D}_{\text{atyp}}$). We evaluate the models' performance on test "typical" set $\mathcal{D}'_{\text{typ}}$ which is defined in Section 3.2.

**(i) Additional Results for Preliminary Study - Section 3.2 In Traditional ERM**

Fig. 14, Fig. 15 and Fig. 16 report the performance of traditional ERM, trained on different resampled datasets with different amount of atypical samples existed. The figures report the clean accuracy on test typical set of CIFAR100, CIFAR10 and Tiny ImageNet. We also leave the robustness performance out here as the models are not robust to adversarial attacks. From the results, we can conclude that in traditional ERM, fitting atypical samples will not degrade the models' performance on typical samples. For example, in CIFAR100 dataset, with 100% atypical samples included (Atypical-100%)., the accuracy on the test typical set is even slightly higher than the model trained without atypical samples (Atypical-0%). This conclusion is consistent for all three datasets and model architectures.

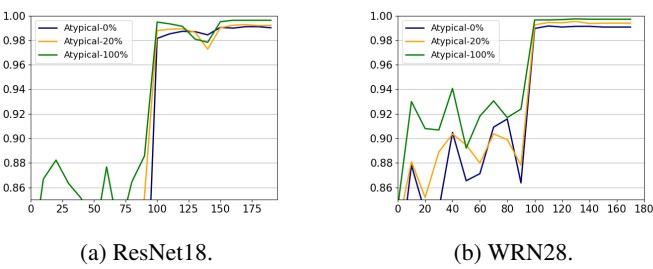

(a) ResNet18.            (b) WRN28.

Figure 14: Clean Accuracy on **Typical** Set of CIFAR100

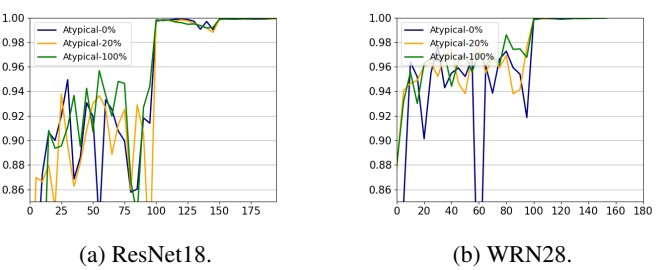

(a) ResNet18.            (b) WRN28.

Figure 15: Clean Accuracy on **Typical** Set of CIFAR10

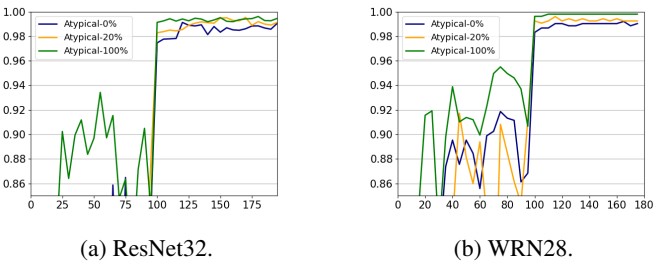

(a) ResNet32.            (b) WRN28.

Figure 16: Clean Accuracy on **Typical** Set of Tiny ImageNet

**(ii) Additional Results for Preliminary Study - Section 3.2 In Adversarial Training**

Fig. 17, Fig. 18 and Fig. 19 report the performance of adversarial training, on different resampled datasets with different amount of atypical samples existed. The figures report both clean and adversarial accuracy on test atypical sets of CIFAR100, CIFAR10 and Tiny ImageNet. Based on the experimental results, we find that including more atypical samples can cause the model have worse performance on typical samples in all three datasets. In datasets with a large portion of atypical samples, such as CIFAR100, the negative effects of atypical samples are more obvious. In CIFAR100, training on datasets with 100% atypical samples (Atypical 100%) can cause the clean & adversarial accuracy drop by $\sim 7\%$ and $8\%$, respectively.

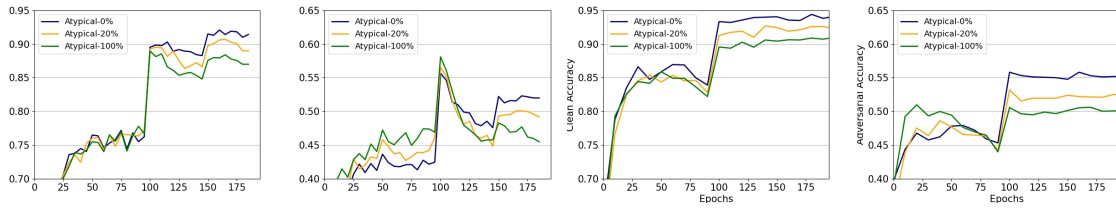

(a) Clean (left) & Adv Acc. (right) under ResNet18. (b) Clean (left) & Adv Acc. (right) under WRN28.

Figure 17: Clean Accuracy and Adversarial Accuracy on **Typical** Set of CIFAR100

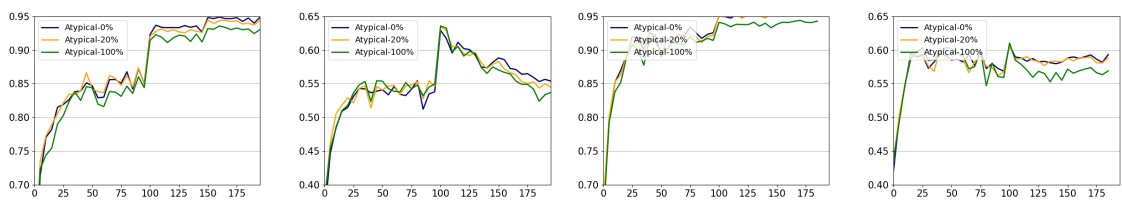

(a) Clean (left) & Adv Acc. (right) under ResNet18. (b) Clean (left) & Adv Acc. (right) under WRN28.

Figure 18: Clean Accuracy and Adversarial Accuracy on **Typical** Set of CIFAR10

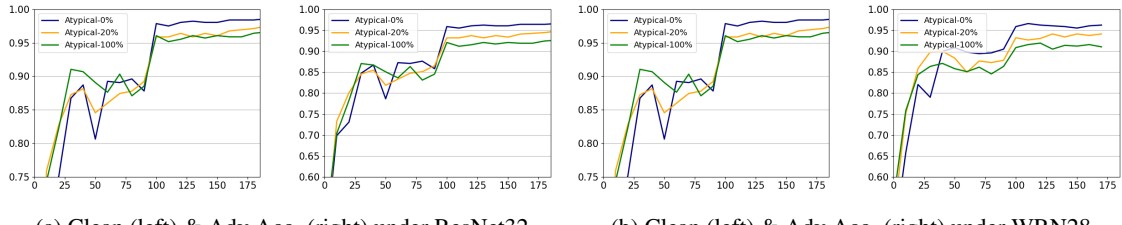

(a) Clean (left) & Adv Acc. (right) under ResNet32. (b) Clean (left) & Adv Acc. (right) under WRN28.

Figure 19: Clean Accuracy and Adversarial Accuracy on **Typical** Set of CIFAR100

### B.3 AN ABLATION STUDY FOR "POISONING ATYPICAL SAMPLES"

In this subsection, we conduct additional experiments to verify the claims about "poisoning atypical samples". In particular, we aim to demonstrate that: the atypical images with high "poisoning scores" are likely to obtain features which visually resemble the images from a (wrong) different class, so fitting them are likely to degrade the model performance. To validate our claim, we print out 200 images of atypical samples from CIFAR100 training set, with highest / lowest poisoning scores as defined in Eq.(6) of the paper. Then, we let two individual human annotators to label each image, by choosing one of 4 options including: (a.) This image belongs to class $y_1$. (b.) This image belongs to class $y_2$. (c.) Both class $y_1$ and $y_2$ are likely. (d.) Neither class $y_1$ or $y_2$. Here, $y_1$ is the ground truth label of the sample; $y_2$ is the class other than $y_1$ which the model has the maximal confidence: $y_2 = arg \max_{t \neq y_1} F_t(x)$, and the model $F$ is obtained via PGD adversarial training, with all atypical samples removed from the training set. We report the percentage of the answers of the human annotators. We report the percentage of the answers of the human annotators.

From the results, we can see, for the samples with high poisoning scores, people are more likely to believe that "the image is from $y_2$" or "the image is both likely from $y_1$ or $y_2$". It is because the highly-poisoning images can persist a lot of similar features from these two classes. It also worth to mention that for samples which people labeled them to $y_1$, they usually also have features belonging to the class $y_2$. For example, a "bowl" which is round and with orange color. Although people believe it is a bowl confidently, it does have features such as "round" and "orange color", which are also features of class "orange".

|  | Class $y_1$ | Class $y_2$ | Both Likely | Neither. | Total |
|---|---|---|---|---|---|
| High Poison Score | 59.0 | 12.0 | 26.0 | 4.0 | 100.0 |
| Low Poison Score | 96.0 | 0.0 | 1.5 | 2.5 | 100.0 |

## C    PROOF OF THEORIES

In Section 3.2, we argue that *the poisoning effect during adversarial training can be significantly stronger than traditional ERM*. In this section, we provide detailed discussions about this claim as well as complete proofs. Our analysis is based on a binary classification task with a two-dimensional Gaussian Mixture distribution:

$$y \overset{u.a.r}{\sim} \{-1, +1\}, \quad \theta = (\eta, 0), \eta \in \mathbb{R}_+, \quad x \sim \begin{cases} \mathcal{N}(\theta, \sigma^2 I) & \text{if } y = +1 \\ \mathcal{N}(-\theta, \sigma^2 I) & \text{if } y = -1, \end{cases} \quad (11)$$

and we consider linear models $F = sign(w^T x + b)$, where $w$ and $b$ are the model weight and bias. Moreover, we assume there is an atypical training sample $x_0 \in \mathbb{R}^2$ labeled as "$-1$", with Euclidean distance to the center of each class $||x_0 - (-\theta)||_2 = d_{-1}, ||x_0 - \theta||_2 = d_{+1}$. Intuitively, this setting of $x_0$ resembles the cases in Figure 3 if $d_{+1}$ is relatively small, since it has a short distance to the wrong class. We also constrain $d_{-1} \leq 2 \times \eta$ for $x_0$ to preserve certain similarity to the class "-1". The following theorem shows that there always exists a model which fits the sample $x_0$, and achieve optimal accuracy (i.e. $> 99\%$) on both classes, although $d_{+1}$ can be small.

**Theorem 1** *For a data distribution in Eq. 4, given that $\eta/\sigma$ is large enough, we define $l = \sqrt{2} \cdot \Phi^{-1}(0.99) \cdot \sigma \cdot (1 + \Phi^{-1}(0.99) \cdot \frac{\sigma}{\eta})^{-1/2}$. For a point $x_0$ with $d_{+1} \leq 2\eta$, if $d_{+1} \geq l$, then there exist a model $F$ fitting $x_0$: $F(x_0) = -1$ and has classification error at most $1\%$ on both classes.*

**Proof 1** *We prove this theorem by two main steps.*

**Step 1.** *we show that for any $x_0$ satisfying $d_{-1} \leq 2\eta$, $d_{+1} \geq l$, at least one of $F_1$, $F_2$ and $F_3$ fits $x_0$: $\exists \, i \in \{1, 2\}, F_i(x_0) = -1$, where:*

$$F_1(x) := sign([1, 0]^T x - (\eta - \Phi^{-1}(0.99) \cdot \sigma))$$
$$F_2(x) := sign([\tau, \sqrt{\eta^2 - \tau^2}]^T \cdot x)$$
$$F_3(x) := sign([\tau, -\sqrt{\eta^2 - \tau^2}]^T \cdot x)$$

*In the following parts, we let $\tau = \Phi^{-1} \cdot \sigma$ for simplicity.* **Step 2.** *Both models $F_1$ and $F_1$ has a classification accuracy at least $99\%$ for both classes.*

**Proof of Step 1.** *For any given sample $x_0 = [p, q]$, on one hand, if $p < \eta - \tau$, $F_1(x_0) = -1$.*

*On the other hand, if $p \geq \eta - \tau$, we will show all samples $x_0 = [p, q]$ which satisfies $F_2([p, q])) \neq -1$, it must have $d_{+1} \leq l$, where $l = \sqrt{2}\sigma \cdot (1 + \tau/\eta)^{-1/2}$. Here, we assume $q \leq 0$ without loss of generality. This can be treated as a quadratic maximization problem under a convex closed bound:*

$$\begin{aligned} \max \;\; & (p - \eta)^2 + q^2 \\ \text{subject to} \;\; & K_1 := \tau \cdot p + \sqrt{\eta^2 - \tau^2} \cdot q \geq 0 \\ & \text{and } K_2 := p \geq \eta - \tau; \\ & \text{and } K_3 := (p - (-\eta))^2 + q^2 \leq (2\eta)^2 \\ & \text{and } K_4 := q \leq 0 \end{aligned} \quad (12)$$

*Figure 20 provides an illustration of the problem proposed in Eq. 12. The optima of Eq. 12 lies on the border of the closed region (shadow area). Next, we solve the values of the object $(p - \eta)^2 + q^2$ on points $S_1$ and $S_2$, and compares the objective value when $x_0 = S_1$ or $x_0 = S_2$.*

*For $x_0 = S_1$, it satisfies:*

$$\tau \cdot p + \sqrt{\eta^2 - \tau^2} \cdot q = 0$$
$$p = \eta - \tau;$$

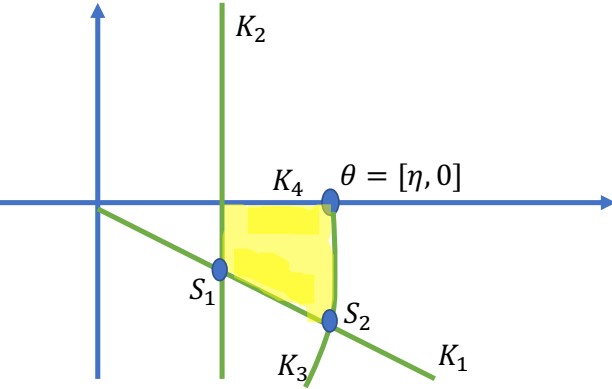

Figure 20: Illustration of Eq. 12

*By solving the equations, we have $(p - \eta)^2 + q^2 = (\sqrt{2} \cdot \tau \cdot (1 + \frac{\tau}{\eta})^{-1/2})^2$.*

*For $x_0 = S_1$, it satisfies:*

$$\tau \cdot p + \sqrt{\eta^2 - \tau^2} \cdot q = 0$$
$$(p - (-\eta))^2 + q^2 = (2\eta)^2$$

*By solving the equations, we have $(p - \eta)^2 + q^2 = (\tau \cdot (1 - \frac{\tau}{\eta} + \Omega((\frac{\tau}{\eta})^2))^{-1/2})^2$. It is not hard to see that, when $\eta/\sigma$ is sufficiently large, the objective will achieve larger value on $S_1$, whose optimum is not larger than $l^2$. Therefore, we finished proving the step 1.*

**Proof of Step 2.** *The distance of the center of class "+1" to the decision boundaries of $F_1$, $F_2$ and $F_3$ are all $\tau$. Therefore, the error of class "+1":*

$$\begin{aligned} Pr.(F(x) \neq +1 | y = +1) &= Pr.(\mathcal{N}(0, \sigma^2) > \tau) \\ &= Pr.(\mathcal{N}(0, 1) > \tau/\sigma) \\ &= Pr.(\mathcal{N}(0, 1) > \Phi^{-1}(0.99)) = 1\% \end{aligned}$$

*For the class "-1", since classifiers $F_2$ and $F_3$ are symmetric, they also have 1% error on class "-1". For classifier $F_1$, it is easy to see that it has smaller error than 1% on class "-1". Based on the results of Step 1 and Step 2, we finished the proof of Theorem 1.*

**Theorem 2** *For the same data distribution in Eq. 4 and same definition $l = \sqrt{2} \cdot \Phi^{-1}(0.99) \cdot \sigma \cdot (1 + \Phi^{-1}(0.99) \cdot \frac{\sigma}{\eta})^{-1/2}$. We assume $l \leq d_{+1} < k \cdot l$, where $1 \leq k << \eta/\sigma$. Then, any model $F$ will have a classification error at least 50% on class "+1", when $F$ fits the adversarial example of $x_0$: $F(x_0 + \delta) = -1, \forall ||\delta||_2 \leq \epsilon$, and*

$$\epsilon \geq \sigma \cdot (k \cdot \sqrt{2} \cdot \Phi^{-1}(0.99)) \cdot (1 + \Phi^{-1}(0.99) \cdot \frac{\sigma}{\eta})^{-1/2} \tag{13}$$

**Proof 2** *During adversarial training, if $\epsilon > k \cdot l$, it will include the center of class "+1" into the adversarial training bound of sample $x_0$:*

$$\theta \in \{x + \delta, ||\delta||_2 \leq \epsilon\}.$$

*Therefore, if there is a model which fits any adversarial counterpart of the sample $x_0$, there does not exist a linear model which has error rate smaller than 50% on class "+1". Therefore, we finished the proof of Theorem 2.*

## D  THE DETAILED TRAINING SCHEME OF BAT

In this section, we provide the detailed training scheme of BAT in Algorithm 1. In particular, BAT algorithm starts from a randomly initialized neural network. On each mini-batch, it applies PGD

attack to generate (training) adversarial examples (Step 5). Following the Eq 6 and Eq 7, BAT calculates which samples are likely to be *Poisoning Atypical Samples* and their corresponding weight values (Step 6). Under the current mini-batch, next, BAT calculates the *Discrimination Loss* of the typical samples $\mathcal{D}_{\text{typ}}$ (Step 8). Finally, BAT uses SGD to update the model parameter to minimize the reweighted adversarial loss regularized by $\beta$ times discrimination loss (Step 8).

---

**Algorithm 1** The Benign Adversarial Training (BAT) Algorithm

---

1: **Input:** Training dataset $\mathcal{D}$, with typical set $\mathcal{D}_{\text{typ}} = \{x \in \mathcal{D}; \text{mem}(x) \leq \sigma\}$, atypical set $\mathcal{D}_{\text{atyp}} = \{x \in \mathcal{D}; \text{mem}(x) > \sigma\}$. Targeted type of adversarial attack: $l_\infty$-$\epsilon$ attack. Hyperparameters $\alpha, \beta \in \mathbb{R}^+$.
2: Randomly initialize the network $F$
3: **repeat**
4:     Fetch mini-batch data $\{(x_i, y_i)\}$ at current epoch
5:     Using PGD to generate adversarial training sample $\{(x_i^{\text{adv}}, y_i)\}$
6:     Calculate Poisoning Score $\mathbf{q}(x_i^{\text{adv}})$ and weight $w_i$ as in Equation 6 and Equation 7.
7:     Calculate Discrimination Loss $\mathcal{L}_{DL}(F)$ within the current mini-batch, as in Equation 9.
8:     Update $F$ by SGD on the objective: $\mathcal{L}_{\text{BAT}} = \frac{1}{\sum_i w_i} \sum_i \left[ w_i \cdot \mathcal{L}(F(x_i^{\text{adv}}), y_i) \right] + \beta \cdot \mathcal{L}_{DL}(F)$.
9: **until** End of Training

---

# E    ADDITIONAL RESULTS FOR EXPERIMENT

In this section, we provide additional experimental results to validate the effectiveness of BAT method. In Table 3 and Table 4, we provide the results of BAT and baseline models on CIFAR100 dataset under ResNet18 and WRN28 architectures. In Table 5 and Table 6, we provide the results of BAT and baseline models on Tiny ImageNet dataset under ResNet32 and WRN28 architectures. In the experiments, we train the models for 160 epochs with learning rate 0.1, momentum 0.9, weight decay 5e-4, and decay the learning rate by 0.1 at the epoch 80 and 120. To have a more comprehensive and reliable adversarial robustness, in addition to PGD adversarial attack Madry et al. (2017), we also measure the model's adversarial accuracy via other attack algorithms, including FGSM attack Goodfellow et al. (2014), CW attack Carlini & Wagner (2017) and Auto Attack Croce & Hein (2020). For PGD attack, we implement the algorithm with 100 steps. For CW and AA attack, we follow the suggestion from the original papers to set the hyperparameters.

Table 3: Performance of BAT vs. Baselines on CIFAR100 Under ResNet18

| Method | Clean Acc. | FGSM | PGD | CW | AA. |
|---|---|---|---|---|---|
| PGD Adv. Training | 56.9 (-2.6) | 36.0 (-1.3) | 27.4 (+0.1) | 25.4 (-1.2) | 23.6 (-0.7) |
| TRADES ($1/\lambda = 5$) | 56.6 (-2.9) | 36.5 (-0.8) | 26.9 (-0.4) | 25.3 (-1.3) | 23.9 (-0.4) |
| MART Wang et al. (2019) | 51.8 (-7.7) | 36.1 (-1.2) | **30.4** (+3.1) | 25.8 (-0.8) | **24.4** (+0.1) |
| GAIRAT Zhang et al. (2020) | 58.2 (-1.3) | 36.5 (-1.2) | 27.8 (+0.5) | 25.9 (-0.7) | 23.8 (-0.2) |
| BAT ($\alpha = 1, \beta = 0.2$) | **59.5** (+0.0) | **37.3** (+0.0) | 27.3 (+0.0) | **26.6** (+0.0) | 24.3 (+0.0) |
| BAT ($\alpha = 2, \beta = 0.2$) | 59.3 (-0.2) | 37.1 (-0.2) | 27.4 (+0.1) | 26.5 (-0.1) | 24.0 (+0.7) |

Table 4: Performance of BAT vs. Baselines on CIFAR100 Under WRN28

| Method | Clean Acc. | FGSM | PGD | CW | AA. |
|---|---|---|---|---|---|
| PGD Adv. Training | 59.7 (-2.3) | 34.9 (-3.7) | 24.7 (-3.8) | 24.2 (-2.3) | 22.5 (-2.3) |
| TRADES ($1/\lambda = 5$) | 57.3 (-4.7) | 34.5 (-4.1) | 24.9 (-3.6) | 24.6 (-1.9) | 22.9 (-1.9) |
| MART (Wang et al., 2019) | 56.5 (-5.5) | 36.1 (-2.5) | 26.8 (-1.7) | 25.3 (-1.2) | 23.8 (-1.0) |
| GAIRAT (Zhang et al., 2020) | 60.2 (-1.8) | 34.8 (-3.8) | 24.4 (-4.1) | 24.8 (-1.5) | 22.9 (-1.9) |
| BAT ($\alpha = 1, \beta = 0.2$) | **62.0** (+0.0) | 38.6 (+0.0) | **28.5** (+0.0) | **26.5** (+0.0) | **24.8** (+0.0) |
| BAT ($\alpha = 2, \beta = 0.2$) | 61.4 (-0.6) | **38.9** (+0.3) | 28.2 (-0.3) | 26.3 (-0.2) | **24.8** (-0.0) |

The results show that the BAT method can consistently outperform baseline models, as BAT has better clean accuracy vs. adversarial accuracy trade-off, especially on the larger model architecture WRN28. Under ResNet18, the BAT method achieves comparable adversarial accuracy to the best

Table 5: Performance of BAT vs. Baselines on Tiny ImageNet Under ResNet32

| Method | Clean Acc. | FGSM | PGD | CW | AA. |
|---|---|---|---|---|---|
| PGD Adv. Training | 56.3 (-3.1) | 37.5 (-2.9) | 32.3 (+0.3) | 29.8 (-1.7) | 29.8 (-2.2) |
| TRADES ($1/\lambda = 5$) | 55.4 (-4.0) | 35.2 (-5.2) | 28.8 (-3.2) | 27.0 (-4.5) | 27.0 (-5.0) |
| MART Wang et al. (2019) | 56.2 (-3.2) | 38.1 (-2.3) | **34.5** (+2.5) | **31.8** (+0.3) | 32.0 (+0.0) |
| GAIRAT Zhang et al. (2020) | 58.4 (-1.0) | 37.3 (-3.1) | 30.4 (-1.6) | 28.9 (-2.6) | 29.0 (-3.0) |
| BAT ($\alpha = 1, \beta = 0.2$) (+0.0) | **59.4** (+0.0) | 40.4 (+0.0) | 32.0 (+0.0) | 31.5 (+0.0) | 32.0 (+0.0) |
| BAT ($\alpha = 2, \beta = 0.2$) | **59.4** (+0.0) | **41.3** (+0.9) | 32.9 (+0.9) | **31.8** (+0.3) | **32.4** (+0.4) |

Table 6: Performance of BAT vs. Baselines on Tiny ImageNet Under WRN28

| Method | Clean Acc. | FGSM | PGD | CW | AA. |
|---|---|---|---|---|---|
| PGD Adv. Training | 58.9 (-3.4) | 35.7 (-5.9) | 31.7 (-4.1) | 30.0 (-3.7) | 30.0 (-4.3) |
| TRADES ($1/\lambda = 5$) | 59.7 (-2.6) | 37.4 (-4.2) | 32.0 (-3.8) | 31.8 (-1.9) | 32.0 (-2.3) |
| MART (Wang et al., 2019) | 58.2 (-4.1) | 41.0 (-0.6) | 35.6 (-0.2) | 33.9 (+0.2) | 34.1 (-0.2) |
| GAIRAT (Zhang et al., 2020) | 60.5 (-1.8) | 39.1 (-2.5) | 33.1 (-2.7) | 31.9 (-1.8) | 32.3 (-2.0) |
| BAT ($\alpha = 1, \beta = 0.2$) | 62.3 (+0.0) | 41.6 (+0.0) | 35.8 (+0.0) | 33.7 (+0.0) | 34.3 (+0.0) |
| BAT ($\alpha = 2, \beta = 0.2$) | **62.4** (+0.1) | **43.1** (+1.5) | **37.4** (+1.6) | **35.3** (+1.6) | **35.6** (+0.7) |

baseline methods, such as MART and GAIRAT, under the strongest attacking methods such as AA attack. However, BATs is around 7.7% higher clean accuracy than MART, and 1.3% GAIRAT. Under a larger model architecture like WRN, BATs have both the highest clean accuracy and adversarial accuracy on various attacking methods. Interestingly, BAT methods take better advantages under large model architectures. The potential reason is that larger neural networks have greater ability to fit those atypical samples during the early stages of training. As a result, the poisoning effect from fitting the "poisoning" atypical samples in baseline methods is stronger. However, BATs can effectively eliminate the poisoning effect and hence enjoy better performance.

# F BOARDER IMPACT

Nowadays, deep neural networks (DNNs) have been widely applied to solve various machine learning tasks, especially on many safety-critical tasks such as autonomous vehicle Fagnant & Kockelman (2015), AI healthcare Jiang et al. (2017) and ID authentication Mohammed et al. (2011), etc. However, the existence of adversarial attacks Xu et al. (2019) brings huge threats to the safety of these DNNs' applications. As one of the most successful approaches to defend DNNs against adversarial attacks, adversarial training methods Madry et al. (2017); Zhang et al. (2016) still suffer from several disadvantages. For example, for a DNN model to achieve good adversarial robustness, it usually sacrifices its clean accuracy. Adversarially trained DNNs also present strong overfitting effects. In our study, we draw important findings to explain these drawbacks of adversarial training from the data perspective and empirically validate these issues' relation to atypical samples in the data distribution. Moreover, the currently existed adversarially trained models can only achieve satisfactory performance on simple datasets such as MNIST and CIFAR10. Motivated from our findings, we propose a new method to improve the performance of adversarial training, especially on more complex datasets. We anticipate that our findings and method are helpful for further studies to improve the effectiveness and feasibility of adversarial training and eventually build safer DNNs.

