# OpenReview forum: "Towards the Memorization Effect of Neural Networks in Adversarial Training"
_ICLR.cc/2022/Conference — ICLR 2022 Submitted_

### Official Review · Reviewer_ytJj · 2021-10-30

**Correctness:** 3
**Technical Novelty And Significance:** 2
**Empirical Novelty And Significance:** 2
**Recommendation:** 5
**Confidence:** 4

**Main Review:**

Strengths:
- The empirical observations on the effect of memorizing atypical samples are interesting.
- The writing is clear, with detailed descriptions of the proposed method.
- It is good to have ablation studies on different modules in BAT.

Weaknesses:
- Since there are several extra modules in the objective of BAT (Eq. (10)), there should be evaluations under adaptive attacks, whose objective involves the terms like $\mathcal{L}\_{DL}$.
- As described in the paper, it seems that the authors do not apply early-stopping during the adversarial training process. This largely weakens the performance of baselines [b]. Besides, more advanced methods listed in [c] should be considered on CIFAR-100.
- CIFAR-100 and Tiny ImageNet are not quite widely used in the adversarial community. It would be more informative to have the results like Table 1 on CIFAR-10, where many benchmarks exist to better estimate the effectiveness of BAT.

Minors:
- As suggested by [a], weak attacks like FGSM should not be treated as an evaluation, at least not in the main text.
- There should be more details on the settings (e.g., iteration steps) of PGD and CW attacks.
- The $1/\lambda$ is 6 in TRADES paper, while it is set as 5 in, e.g., Table 1.

References:
[a] On Evaluating Adversarial Robustness.
[b] Overfitting in adversarially robust deep learning.
[c] RobustBench: a standardized adversarial robustness benchmark.

**Summary Of The Paper:**

This paper investigates the memorization effects in adversarial training, especially on the effect of memorizing atypical examples. Based on the empirical observations, the authors propose benign adversarial training (BAT), which is evaluated on CIFAR-100 and Tiny ImageNet.

**Summary Of The Review:**

Interesting observations on the memorization effects of atypical samples in adversarial training, but the empirical evaluations should involve stronger baselines (e.g., using early-stop) and adaptive attacks (e.g., considering the BAT modules), while it would be much more informative to have, e.g., AA results on CIFAR-10.

---

> ### Author Response · Authors · 2021-11-22
> **Response to Reviewer ytJj**
>
>
> Thank you for reviewing our paper and we are happy to read that "the observations are interesting" and "the paper is well-written". In the following, we will clarify your key concerns. Based on the suggestions of the reviewer, we will provide additional results including: (1) the performance of BAT on adaptive attacks; (2) clarification on baseline methods; (3) performance of BAT on CIFAR10. Next, we will answer these questions and provide additional results.
>
> **(1) Adaptive Attacks on BAT**
>
> We follow the suggestion of the reviewer, we devise adaptive attack method to further evaluate the robustness of BAT in CIFAR100, by taking account of the $L_{DS}$ term. In particular, we modify the objective of PGD attack, by searching (adversarial) examples $x'$ around each clean test sample $x$ (with true label $y$), to have maximal (weighted) sum of cross-entropy loss and "discrimination loss":
> $$\max_{||x'-x||\leq\epsilon} (1-\beta)\cdot  L_{CE}(x', y) + \beta \cdot L_{DL}(x', t, y) \text{~~~} \forall t\neq y.$$
> $$L_{DL}(x', t, y) = \sum_{j = 1}^J\left[- \log ~~\frac{e^{h(x') \cdot h(x_j) / \tau}}{ \sum_{k=1}^K  e^{h(x') \cdot h(x_k) / \tau}} \right]$$
> $$\text{where  } y_j = y, y_k = t, \forall j = 1.2,...J; k = 1.2,...K$$
>
> Note that the original "discrimination loss" of this paper is defined on a mini-batch of training samples. Thus, here we randomly samples training samples from the true class  $\{y_j\}$ , as well as training samples from each single wrong class $\{y_j=k\}_{k = 1}^K$. In this way. the DL loss here encourage the algorithm to search sample $x'$ which has distinct representation to its true class, but has similar representation to a wrong class. We also set different $\beta$ to balance between cross entropy loss and discrimination loss. To optimize the objective above, we apply the PGD algorithm with 100 steps for each sample and step size $2/255$.
>
> |           | Clean Acc | PGD  | Adapt, b=0.5 | Adapt, b=1.0 |
> |-----------|-----------|------|--------------|--------------|
> | BAT, a= 1 | 62.0      | 28.5 | 28.4         | 28.6         |
> | BAT, a= 2 | 61.4      | 28.2 | 28.0         | 28.0         |
>
> From the results, we can see the adaptive attack does not significantly have better attacking performance than original PGD method. It may be because our proposed method didn't involves issues such as gradient masking [1], so PGD is a reliable method to evaluate the robustness of our model. Moreover, we also report the results of AA in the original paper. Since it involves black box attacks which also overcome gradient masking issues, AA attack is a reliable evaluation method.
>
> **(2) Comparison of BAT against Baselines**
>
> We do consider apply early-stopping strategy during the adversarial training process. In Section 5.1 of the original paper, we clearly states that the performance of baseline methods are reported on the checkpoints with the best PGD adversarial accuracy. Our reported numbers are consistent to the results listed in the RobustBench Leadingboards [2].
>
> Admittedly, there are several methods listed in [2] achieving higher performance than BAT (i.e., on CIFAR100). However, the two components of BAT, reweighting and DL loss, are both orthogonal to those methods. Thus, adopting BAT to the top-ranked methods may further boost the performance of their methods. Remind that in our paper, we compared our methods those SOTA methods which also consider reweight, such as MART and GAIRAT and showed BAT has consistent better performance.
>
> **(3) Performance of BAT on CIFAR10**
>
> We follow the suggestion of the reviewer, providing additional experimental results on CIFAR10 dataset under WRN28-10 architecture. In order to evaluate their performance, we compare their clean accuracy as well as adversarial accuracy by PGD attack and Auto-Attack. Similar to the results in the paper, we also report the checkpoints when the models achieve the best PGD adversarial accuracy.
>
> |           | Clean Acc  | PGD        | AA         |
> |-----------|------------|------------|------------|
> | PGD Train | 85.8(-0.6) | 53.9(-0.3) | 51.9(-0.2) |
> | TRADES    | 84.6(-1.8) | 54.5(+0.3) | 52.5(+0.1) |
> | MART      | 83.5(-2.9) | 56.5(+2.3) | 54.3(+2.2) |
> | GAIRAT    | 87.8(+1.4) | 54.0(-0.2) | 52.2(+0.1) |
> | BAT       | 86.4(+0.0) | 54.2(+0.0) | 52.1(+0.0) |
>
> From the results, we can see BAT can still (slightly) outperform PGD adversarial training and has better clean accuracy than TRADES. Moreover, BAT has higher clean accuracy than MART but has lower adversarial accuracy.
> Notably, BAT is granted to have better advantage in complex datasets containing large portion of atypical samples, such as CIFAR100 and Tiny~ImageNet which contains $>40\%$ and $>49\%$ atypical samples respectively. Therefore, BAT can always have better performance in these datasets than a relatively simpler dataset such as CIFAR10.

---

> > ### Author Response · Authors · 2021-11-22
> > **Response to Reviewer ytJj  (Part-2)**
> >
> > **(4) Additional Clarifications on Minor Concerns**
> >
> > 1. In addition to FGSM, our experimental results also consider strong attacks, including PGD, CW and Auto Attack.
> >
> > 2. Following the suggestion of the reviewer, we include more detailed descriptions of the robustness evaluation methods, such as PGD and CW, in the rebuttal revised version.
> >
> > 3. We re-implement the setting of the baseline method TRADES, with $1/\lambda = 6$ for CIFAR100 dataset under WRN28. From the table below, we could get similar results when $1/\lambda = 5$, and BAT have consistent advantage over TRADES in CIFAR100.
> >
> > |                       | Clean Acc. | PGD  | AA   |
> > |-----------------------|------------|------|------|
> > | TRADES($1/\lambda=5$) | 57.3       | 24.9 | 22.9 |
> > | TRADES($1/\lambda=6$) | 57.1       | 25.3 | 23.1 |
> >
> > **Reference**
> >
> > [1] Obfuscated gradients give a false sense of security, Athalye et al, 2017
> >
> > [2] RobustBench: a standardized adversarial robustness benchmark. Croce et al, 2020

---

> ### Author Response · Authors · 2021-11-25
> **About the Rebuttal to Reviewer Response**
>
> Hi, we sincerely appreciate your review. Since we are approaching the end of the discussion period, we would like to enquire if the reviewer has any remaining questions or further points to discuss after our response?

---

> ### Author Response · Authors · 2021-11-29
> **Final-Stage Discussion About the Paper**
>
> Hi, dear reviewer,
>
> We hope our responses can address your main concerns about our paper. In particular, we believe that our additional clarification below can successfully validate the rigorousness of our analysis \& evaluation. Please shoot any remaining questions, as we are approaching the end of the discussion stage. Thanks.

---

### Official Review · Reviewer_iXiX · 2021-10-30

**Correctness:** 3
**Technical Novelty And Significance:** 2
**Empirical Novelty And Significance:** 2
**Recommendation:** 5
**Confidence:** 4

**Main Review:**

Pros:

1. This paper makes a connection between the memorization effect of DNN with adversarial training and provides some comprehensive experimental results on analyzing the effect of training data for adversarial robustness. And this study shows a promising perspective for adversarial robustness, which focuses on the data perspective.
2. The proposed benign adversarial training gains consistent improvement (at least using the two benchmark datasets) on both natural and robust performance across different evaluations (including the strong attack method---AA).

Cons:

1. The major concern is the novelty of the main contributions. The similar "important" findings in this paper have already been pointed out and analyzed by previous studies [1,2,3]. For example, in [1], the authors also investigated memorization behavior in adversarial training and have pointed out that overfitting some rare examples in training can just improve the clean accuracy, and ignoring these rare examples helps in adversarial robustness. And in [3], the authors show that using high-quality (not so rare or hard) examples can indeed improve both natural performance and adversarial robustness, which has a consistent high-level idea with benign adversarial training.
2. Some claims are not well-supported. It may be better to add some citations to justify or provide some empirical results, especially for the memorization behavior in adversarial training. For example, how to confirm that these atypical samples have deviated from the main sub-population? Whether DNN can indeed fit every atypical samples' labels since we know that the model capacity for adversarial training is always not enough (in other words, adversarial training may require DNNs with a large model capacity than standard training [4]).
3. The motivation for exploring or connecting memorization with adversarial training is not clear and not strong (At the top of page 2). The recent study mainly focuses on the "overfitting" phenomena in adversarial training, so what's the relationship of memorization in adversarial training with such "overfitting" phenomena? (Personally, since there seems to be a logic gap, it is hard to understand why we need to explore the memorization behavior in adversarial training).
4. The reweighting strategy may need further explanation or illustration.
5. What's the relationship of the proposed method with adopting curriculum learning in adversarial training?
6. Empirically, if the data distribution is long-tail, how can we apply the benign adversarial training? Can benign adversarial training also effectively utilize the long-tailed data since it seems to be similar to the atypical samples mentioned in this paper?

Minor:

It is better to use $f$ instead of $F$ in all the equations to appropriately indicate the loss function.

References:

[1] Sanyal, Amartya, Puneet K. Dokania, Varun Kanade, and Philip Torr. "How Benign is Benign Overfitting?." In *International Conference on Learning Representations*. 2020.

[2] Dong, Yinpeng, et al. "Exploring Memorization in Adversarial Training." *arXiv preprint arXiv:2106.01606* (2021).

[3] Dong, Chengyu, Liyuan Liu, and Jingbo Shang. "Data Profiling for Adversarial Training: On the Ruin of Problematic Data." arXiv preprint arXiv:2102.07437 (2021).

[4] Wu, Boxi, et al. "Do Wider Neural Networks Really Help Adversarial Robustness?." *arXiv preprint arXiv:2010.01279* (2020).

**Summary Of The Paper:**

This paper explores the "memorization" in adversarial training. Through some empirical experiments and theoretical analysis, this paper points out two findings: (a) memorizing atypical samples can improve DNN’s accuracy on clean atypical samples, but hardly improve their adversarial robustness and (b) memorizing some atypical samples can even hurt the DNN’s performance on typical samples. Based on these findings, the authors propose a method called benign adversarial training by reweighting. Empirical results show that it can achieve better clean accuracy or robustness than some baseline methods in CIFAR-100 and Tiny-ImageNet.

**Summary Of The Review:**

Overall, this paper explores the memorization effect in adversarial training and proposes benign adversarial training which further enhances adversarial training. However, although the direction is promising, the major concern is about the novelty of the main contribution since the similar findings in this paper have already been pointed out and analyzed in previous studies [1,2,3], and simply dropping some called "atypical" data can also improve adversarial training [3]. Considering the similar results from the perspective of data quality for adversarial robustness,  the current submission may need some further exploration to provide some different and in-depth findings.

---

> ### Author Response · Authors · 2021-11-22
> **Response to Reviewer iXiX**
>
> Thank you for reviewing our paper. In the following, we will clarify your key concerns.
>
> **Q1. What is the major difference of our paper over the papers as the reviewer mentions, including [1-3]?**
>
> === Our paper vs. Sanyal et al [1] ===
>
> The paper Sanyal [1] suggests: one possible reason of DNNs' vulnerability to adversarial attacks might be because of DNN's ability to fit (memorize) noisy labeled samples. However, we find that our paper has very different and even opposite observations. Moreover, the problem studied in our paper is more general for most of benchmark image datasets. Next, we will detail these differences.
>
> **1.** In Section 3.2 of Sanyal et al [1], the authors claim that the robust trained DNN models always avoid fitting atypical samples. However, the results in our paper show that it is not true. Our paper shows that the robust trained DNN models (on the representative architecture WRN28), can fit all (adversarial) atypical samples. Based on Section 3.1 \& Appendix B.1 of our paper, the models can achieve 100\% training adversarial accuracy on these atypical samples for datasets CIFAR10, CIFAR100 and TinyImageNet. This finding is totally opposite to the observation in Sanyal et al [1].
>
> **2.** In the theoretical results (Section 2.1) of Sanyal et al [1], the authors focus on natural training, and claim fitting "noisy-labeled" samples can cause adversarial vulnerability. However, we claim fitting "poisoning" atypical samples in adversarial training will hurt both clean accuracy and adversarial robustness (of typical samples). For example, in Figure 2 (Section 3.2) of our paper, we show that removing atypical samples will help improve both clean accuracy \& adversarial accuracy of typical samples. In Theory 2 of our paper, we show the model will have both large clean \& adversarial error on the class "+1". In another word, our paper also provides insightful understanding about accuracy vs. robustness trade-off [4].
>
> More importantly, in Theorem 1 of our paper, we show that fitting the (same) "poisoning" atypical samples will not hurt the accuracy of a naturally trained model. As stated in Theorem 1, if an atypical sample $x$ has a distance to class $+1$ larger than $l$, there will exist an optimal classifier with classification error at most $1\%$. In another word, these "poisoning" atypical samples are harmful in adversarial training, but they are not necessarily harmful in natural training. It highlights the key difference between the behavior of natural training \& adversarial training.
>
> **3.** Our paper discusses the impacts of DNN's ability to fit / memorize "atypical" samples, which appear much more frequently than the "noisy-labeled" samples as discussed in Sanyal etal [1]. For example, we show that all common benchmark datasets, CIFAR10, CIFAR100 and TinyImageNet have a large portion of atypical samples, which are at least 11\%, 40\% and 49\%. While, there are only a few wrongly labeled samples in these datasets. Therefore, the analysis of our paper has greater importance to understand the phenomenon of adversarial robustness in general cases / datasets where the training data is correctly labeled.
>
> === This paper vs. Dong et al [2] ===
>
> Admittedly, the paper Dong et al [2] shows some similar findings as our paper, i.e, (a.) adversarially trained DNN models have sufficient capacity to fit / memorize the whole training dataset, even though there is a large portion of "noisy-labeled" data. (b.) fitting some "hard" examples might hurt the performance of adversarial training. Our paper conducts a more rigours study about the harmful effect of fitting certain atypical samples. We also build a strict theoretical model to show fitting these atypical samples will not bring harmful effect in natural training, which highlight the key difference between the behavior of natural training vs. adv. training. Furthermore, our paper proposed a total different method which achieves comparable performance with Dong [2].  Remarkably, our paper and Dong et al [2] are preprinted / submitted to ICLR at the similar time.
>
> === This paper vs. Dong et al [3] ===
>
> The paper Dong et al [3] suggests that there are samples with "bad quality" in common image datasets. Thus, fitting these "bad quality" samples will bring negative effects to the performance of adversarial training. Although our paper also claims that fitting certain atypical samples will degrade the performance of the model, we didn't claim that they are "bad quality" data. For example, in natural training, fitting the atypical samples will boost the test performance of the model. Moreover, our proposed method, BAT, can eliminate the negative effect from harmful atypical samples, but also preserve the model's ability to fit "benign" atypical samples. Given that there is usually a large portion of atypical samples in common datasets, our method is granted to take advantage over simply removing all of them, which is proposed by Dong et al [3].

---

> > ### Author Response · Authors · 2021-11-22
> > **Response to Reviewer iXiX (Part-2)**
> >
> > **Q2. (a.) How to confirm that atypical samples have deviated from the main sub-population? (b.) Whether DNN can indeed fit every atypical samples in adversarial training?**
> >
> > (a.) In Figure 5 of our paper, we show several cases to demonstrate atypical samples have visually very different features with the images that are typical. Moreover, the work Carlini et al [5] also deploys experiments to confirm whether the atypical samples found by the Leave-out method (same as in our paper) truly deviate from the main sub-population. For example, they found that the atypical samples (found by leave-out method) are also given disagreed prediction results, or lower prediction confidence, by various DNN models with distinct architectures and initializations. The results show that the atypical samples we found are truly deviated from the main distribution and they are hard to be represented by samples in the main distribution.
> >
> > (b.) Our paper clearly shows that the robust trained DNN models (under the architecture WRN28), can fit all (adversarial) atypical samples. From the results on Section 3.1 \& Appendix B.1 of our paper, the models can achieve 100\% training adversarial accuracy on these atypical samples for datasets CIFAR10, CIFAR100 and TinyImageNet. It suggests the model have already memorize thee whole training set, as well as their adversarial counterparts.
> >
> > **Q3. Why do we specifically study "atypical samples"?**
> >
> > The main scope of this paper is similar to the previous papers [6-8] which studies the overfitting (generalization) problems in natural training.
> > In particular, the paper [6] reveals that (overparameterized) DNNs have sufficient capacity to memorize the entire training set, even including many outliers or noisy labeled samples. The papers [7-8] suggest that this memorization property of DNNs can benefit their generalization performance, as they can memorize "atypical" samples and correctly predict them during test. However, shallow models cannot memorize "atypical" samples and fail to predict them during test.
> >
> > In our paper, we study the similar problem of "memorization", but focused on adversarial training. We reveal key clues to understand robust overfitting issues through the study about atypical samples. For example, we disclose the fact that memorizing atypical samples during adv. training cannot as effectively improve the generalization performance (as in natural training) of robust DNNs (Section 3.1). We demonstrate that memorizing atypical samples will even bring negative effect to the model's generalization performance (Section 3.2). While, this negative effect does not exist in natural training. Our analysis sheds light on the intrinsic difference between natural training and adv. training, in terms of their generalization behavior.

---

> > > ### Author Response · Authors · 2021-11-22
> > > **Response to Reviewer iXiX (Part-3)**
> > >
> > > **Q4. More Clarification on the Reweighting Method**
> > >
> > > In Section 4, we propose the Reweight strategy to down-weight the importance of "poisoning atypical samples" during training. The "poisoning atypical samples" refer to atypical samples which are very close to the main-subpopulation of a wrong class. In Theory 2 of our paper, we show that fitting the atypical samples which are very close the main distribution of  another class will degrade the model's clean \& adversarial performance. Therefore, our method can effectively improve the model performance by reducing the weights of poisoning atypical samples.
> > >
> > > Moreover, in Appendix B.3 of the rebuttal-revised version (also in the response to Reviewer sm19), we add an additional experiment (including human annotation) to further verify that: the atypical samples with high poisoning scores are visually very close to the main sub-population of a wrong class. This experiments further support our claims about "poisoning" atypical samples and our proposed reweight strategy.
> > >
> > > **Q5. What is the relation between the proposed Reweight method to Curriculum Learning**
> > >
> > > Curriculum learning defers the model to fit the "hard examples" later than "easy" examples. While, our method prevents the model to fit "poisoning atypical samples". Admittedly, "poisoning" atypical samples are deviated from the main-subpopulation, so they are relatively harder to learn. However, hard samples are not "poisoning" atypical samples. They could be benign atypical samples, which deviate from their main sub-population, but are not close to any other classes. Our proposed method can preserve the model's ability to fit benign atypical samples.
> > >
> > > **Q6. If the data distribution is long-tail, how to apply the BAT method?**
> > >
> > > Our proposed method BAT assumes the training data for each class has sufficient samples to reflect data distribution. Otherwise, it is impossible to distinguish between typical \& atypical samples.
> > >
> > > **References:**
> > >
> > > [1] How Benign is Benign Overfitting? Sanyal et al, 2020.
> > >
> > > [2] Exploring Memorization in Adversarial Training, Dong et al, 2021.
> > >
> > > [3] Data Profiling for Adversarial Training: On the Ruin of Problematic Data, Dong et al, 2021.
> > >
> > > [4] Robustness May Be at Odds with Accuracy, Tsipras, 2018
> > >
> > > [5] Distribution density, tails, and outliers in
> > > machine learning: Metrics and applications. Carlini et al, 2019
> > >
> > > [6] Understanding Deep Learning Requires Rethinking Generalization, Zhang et al, 2017
> > >
> > > [7] Benign Overfitting in Linear Regression, Bartlett et al, 2019
> > >
> > > [8] What Neural Networks Memorize and Why: Discovering the Long Tail via Influence Estimation, Feldman et al, 2020

---

> ### Author Response · Authors · 2021-11-25
> **About the Rebuttal to Reviewer Response**
>
> Hi, we sincerely appreciate your review. Since we are approaching the end of the discussion period, we would like to enquire if the reviewer has any remaining questions or further points to discuss after our response?

---

> > ### Comment · Reviewer_iXiX · 2021-11-29
> > **Thanks for your detailed response!**
> >
> > I appreciate the authors for the detailed response, as well as some additional clarification on the raised questions. I think some of them are clearly explained. Nevertheless, the major concern about the novelty of this work is still remaining. Those difference from the previous papers seems not significant, I think this work is the further discussion about fine-grained understanding of AT from the perspective of fitting training examples.

---

> > > ### Author Response · Authors · 2021-11-30
> > > **Thanks again for your valuable review.**
> > >
> > > Thanks again for your valuable review. We believe our paper uncovers important findings, which are not exploited by previous works. Different from the papers listed by the reviewer, our discussion about “atypical samples” highlights the key difference between the behavior of adversarial training and natural training. Specifically, fitting atypical samples in natural training is beneficial to the model’s test accuracy, but fitting (some certain) atypical samples in adversarial training can be harmful to both clean accuracy \& adv. robustness. As a summary, our work provides deep insights to better understand robustness generalization issues, as well as a novel and effective defense algorithm BAT. Again we sincerely appreciate your valuable feedback.

---

### Official Review · Reviewer_sm19 · 2021-11-02

**Correctness:** 4
**Technical Novelty And Significance:** 3
**Empirical Novelty And Significance:** 3
**Recommendation:** 8
**Confidence:** 4

**Main Review:**

The premise of the paper is interesting. While previous work has discussed "atypical examples" in the context of adversarial training, this is (to the best of my knowledge) the first work that performs such a study in a rigorous manner by considering proper measures of atypicality through influences. The findings are thus not always surprising given prior work, but are nevertheless interesting and the proposed method does show promise.

There are however a few issues that I would like to point out:

- The theoretical setting is actually quite similar to the one in [Sanyal et al.]([https://arxiv.org/abs/2007.04028](https://arxiv.org/abs/2007.04028)) (cited). Their focus is on label noise instead of atypical examples (which is not a very different setting after all), but both analyses boil down to robustly classifying samples from a Gaussian in the presence of outliers. This does diminish the impact of the paper somewhat. In any case, the discussion of that paper in the manuscript should be expanded.
- The discussion around "poisoning atypical examples" (e.g., Figure 3) is intriguing but somewhat speculative. Is there a more rigorous way to explore this phenomenon? E.g., by choosing samples with high influence and verifying them through human annotation?
- It is not entirely clear why the discrimination loss regularization would be beneficial in this setting. The authors do provide a justification based on the class margin in representation space. However, since the model is trained end-to-end, why wouldn't standard adversarial training discover this solution?

Minor comments (not affecting score):

- The title is confusing, what does "towards the ... effect" mean?
- [Balaji et al. 2019]([https://arxiv.org/abs/1910.08051](https://arxiv.org/abs/1910.08051)) is quite relevant and worth discussing.
- Why are the PGD-generated adversarial examples during adversarial training referred to as "manually generated"?
- Figure 4 is hard to parse in 3-D. It seems that a table would provide the same information in a more readable format.

**Summary Of The Paper:**

The authors study the interaction between adversarial training (meant to produce robust models) and atypical examples in standard datasets (examples where generalization is driven by a handful of examples, cf [Feldman 2020]). They find that, in contrast to standard training, adversarial training has a hard time generalizing to atypical test examples in the adversarial setting. They also find that adding atypical examples to adversarial training can actually hurt test accuracy, in contrast to standard training where such examples typically help. Finally, the authors propose a method that down-weights  atypical examples during robust training and introduces a contrastive regularizer, which they term BAT (benign adversarial training).

**Summary Of The Review:**

Overall, despite its shortcomings, the paper presents some findings that would be of interest to the community.

---

Post-rebuttal update: Given the authors' clarification of their theoretical contributions and the provided small-scale user study, I increase my score.

---

> ### Author Response · Authors · 2021-11-22
> **Response to Reviewer sm19**
>
> Thank you for reviewing our paper and we are happy to read that "the premise and observations are interesting" and "the analysis is rigorous". In the following, we will clarify your key concerns. Based on the suggestions of the reviewer, we will discuss: **(1)** What is the  differences theoretical results of this paper and Sanyal et al. **(2)** Clarifications of "poisoning atypical samples" **(3)** Clarifications of discrimination loss.
>
> **Q1. What is the difference of the theoretical result of this papaer with Sanyal et al?**
>
> In the theoretical results (Section 2.1) of Sanyal et al [1], the authors focus on natural training, and claim fitting "noisy-labeled" samples can cause adversarial vulnerability. However, our paper is focused on adversarial training and we claim fitting "poisoning" atypical samples in adversarial training will hurt both clean accuracy and adversarial robustness (of typical samples) (Theorem 2).
>
> More importantly, in Theorem 1 of our paper, we show that fitting the (same) "poisoning" atypical samples will not hurt the accuracy of a naturally trained model. As stated in Theorem 1, if an atypical sample $x$ has a distance to class $+1$ larger than $l$, there will exist an optimal classifier with classification error at most $1\%$. In another word, these "poisoning" atypical samples are harmful in adversarial training, but they are not necessarily harmful in natural training. It highlights the key difference between the behavior of natural training \& adversarial training.
>
> In addition to the theoretical results, our paper also have very different findings with Sanyal et al [1]. For example, Sanyal et al [1] claims that the robust trained DNN models always avoid fitting atypical samples. However, the results in our paper show that it is not true. Our paper shows that the robust trained DNN models (under the architecture WRN28), can fit all (adversarial) atypical samples. Moreover. our paper discusses the impacts of DNN's ability to fit / memorize "atypical" samples, which appear much more frequently than the "noisy-labeled" samples as discussed in Sanyal et al [1]. It furthur stress the importance of the finding in our paper.
>
> **Q2. More rigorous way to discuss "poisoning" atypical samples?**
>
> We follow the suggestion of the reviewer, conducting additional experiments to verify the claims about "poisoning atypical samples". In particular, we aim to demonstrate that: the atypical images with high "poisoning scores" are likely to obtain features which visually resemble the images from a (wrong) different class, so fitting them are likely to degrade the model performance.
>
> To validate our claim, we print out 200 images of atypical samples from CIFAR100 training set, with highest / lowest poisoning scores as defined in Eq.(6) of the paper. Then, we let two individual human annotators to label each image, by choosing one of 4 options including: (a.) This image belongs to class $y_1$. (b.) This image belongs to class $y_2$. (c.) Both class $y_1$ and $y_2$ are likely. (d.) Neither class $y_1$ or $y_2$. Here, $y_1$ is the ground truth label of the sample; $y_2$ is the class other than $y_1$ which the model has the maximal confidence:  $y_2 = arg\max_{t\neq y_1} F_t(x)$, and the model $F$ is obtained via PGD adversarial training, with all atypical samples removed from the training set. We report the percentage of the answers of the human annotators.
>
> |                   | Class y1 | Class y2 | Both Likely | Neither. | Total |
> |-------------------|----------|----------|-------------|----------|-------|
> | High Poison Score | 59.0     | 12.0     | 26.0        | 4.0      | 100.0 |
> | Low Poison Score  | 96.0     | 0.0      | 1.5         | 2.5      | 100.0 |
>
> From the results, we can see, for the samples with high poisoning scores, people are more likely to believe that "the image is from $y_2$" or "the image is both likely from $y_1$ or $y_2$". It is because the highly-poisoning images can persist a lot of similar features from these two classes. It also worth to mention that for samples which people labeled them to $y_1$, they usually also have features belonging to the class $y_2$. For example, a "bowl" which is round and with orange color. Although people believe it is a bowl confidently, it does have features such as "round" and "orange color", which are also features of class "orange". In Appendix rebuttal-revised version of the paper, we added an additional subsection about this experiment.
> Due to the time limit, we only have 2 participants taking the survey. We will continue to involve more participants.

---

> > ### Author Response · Authors · 2021-11-22
> > **Response to Reviewer sm19 (Part-2)**
> >
> > **Q3. What is the benefit of "discrimination" loss?**
> >
> > As the reviewer mentioned, end-to-end trained DNN models usually render well-discriminated representations for different class in pen-ultimate layer. However, it is not always true for adversarial training models. For example, the paper [2] empirically finds that adversarially trained models usually have closer proximity of different class samples in the learned feature space. The paper [3] finds adversarial training models are much harder (requiring much more iterations) to find larger-margin data separators. In our paper, we hypothesize the existence of atypical samples (especially poisoning atypical samples) is one critical reason to lead the model have more "mixed" representations for different classes.
> > Therefore, we are motivated to explicitly regularize the discrimination between the typical samples from different classes.
> >
> > Moreover, using $L_{DS}$ loss in Eq.(9) helps the model render compacted representations only for typical samples in one class. It means that the model is not required to output similar representations for those "atypical" samples in this class. In fact, these atypical samples have (visually) very different features to the typical samples in each class (Figure 5 in the paper). Therefore, our proposed method is also encouraged to output representations which better align with human vision. Similar to the response to Reviewer qCTN, we conduct an experiment to further show the benefit of $L_{DS}$. We compare BAT with $L_{DS}$ only regularizing the representations of typical samples (DS), vs. BAT with $L_{DS}$ regularizing the representations of all samples (DS*). Note that we don't apply the reweighting strategy of BAT here. From the result below, we can see that BAT-DS method outperform BAT-DS*, which validate the effectiveness of design of $L_{DS}$.
> >
> > |         | Clean Acc. | PGD Adv. Acc |
> > |---------|------------|--------------|
> > | BAT-DS  | 61.9       | 27.7         |
> > | BAT-DS* | 60.1       | 25.3         |
> >
> > **Q4. Difference to Balaji et al?**
> >
> > The paper Balaji et al [4] claims that the harmful samples (for adversarial training) are samples closer to the decision boundary / closer to other samples in the data input space. Our paper states the harmful (atypical) samples persist similar features to other (typical) samples. In another word, the harmful (atypical) samples are closer to typical samples in the data distribution, instead of data space. Moreover, the paper [5] empirically demonstrates that the images from benchmark datasets are far away enough in the input space.
> >
> > **Reference**
> >
> > [1] How Benign is Benign Overfitting? Sanyal et al, 2020
> >
> > [2] Adversarial Defense by Restricting the Hidden Space of Deep Neural Networks, Mustafa et al, 2019
> >
> > [3] Convergence and Margin of Adversarial Training on Separable Data, Charles et al 2019
> >
> > [4] INSTANCE ADAPTIVE ADVERSARIAL TRAINING:
> > IMPROVED ACCURACY TRADEOFFS IN NEURAL NETS, Balaji et al, 2019
> >
> > [5] A Closer Look at Accuracy vs. Robustness, Yang et al, 2020

---

> > > ### Comment · Reviewer_sm19 · 2021-11-25
> > > **Rebuttal response**
> > >
> > > I appreciate the authors rebuttal.
> > >
> > > - Thank you for clarifying the differences to Sanyal et al. I do now see that while the setting is the same (outliers in Gaussian classification) the analysis and results are quite different.
> > > - The additional analysis of "poisoning atypical examples" is actually insightful and a step in the right direction. I understand that it is limited due to time constraints but I would encourage the authors to include a rigorous study of a similar character in the updated manuscript.
> > > - Unfortunately, I am still not sure that all the components of BAT are fully justified and relevant to atypical examples, even given the provided ablation. I do however understand that it is not always easy to fully justify the effectiveness of a method, especially when the performance improvement is rather small.
> > >
> > > Thus, given the update, I will raise my score to a 7.

---

> > > > ### Author Response · Authors · 2021-12-03
> > > > **Thanks for the Responses**
> > > >
> > > > We gratefully appreciate your time in reviewing our paper, your insightful comments and your support! Your positive feedback is very encouraging for us.

---

> ### Author Response · Authors · 2021-11-25
> **About the Rebuttal to Reviewer Response**
>
> Hi, we sincerely appreciate your review. Since we are approaching the end of the discussion period, we would like to enquire if the reviewer has any remaining questions or further points to discuss after our response?

---

### Official Review · Reviewer_qCTN · 2021-11-03

**Correctness:** 3
**Technical Novelty And Significance:** 3
**Empirical Novelty And Significance:** 3
**Recommendation:** 6
**Confidence:** 3

**Main Review:**

Strengths:
- Interesting observations and analysis provided on the behavior of atypical samples in the context of robust overfitting
- Promising empirical results showing improved clean accuracy at similar or better levels of robustness with strong attacks
- Analysis of the effect of individual components in ablation studies
- Paper is well-written and easy to follow

Weaknesses:
- The choice of threshold for mem(x) used to define an atypical sample seems arbitrary. How do the results vary with different thresholds for mem(x) used to define the atypical set?
- Even though the paper is initially motivated by atypical samples, the mechanism relating atypical-ness to adversarial robustness is not clear and moreover it seems to me that the analysis in "Poisoning Atypical Samples" and the algorithm design that follows can be better understood from the perspective of margins and adversarial robustness; examples with smaller margins (closer to the decision boundary) are necessarily more susceptible to adversarial attack. In particular:
  * Theorems 1/2 are not specific to the notion of atypical samples, and are about samples near the decision boundary (low margin). These theorems are essentially saying that by trying to accomodate these samples near the decision boundary for an optimal classifier, the model will be less robust.
  * The two components of BAT involve other elements that are not related to atypicalness but rather margin - the reweighting involves a poisoning score that is essentially the margin of the example ("likely to be close to the distribution of a wrong class"), and the discrimination loss may be viewed as attempts to increase the margin between the different classes.

  It would be useful to isolate the effect and importance of the atypical-ness of an example in the algorithm, for instance by applying the algorithm to all examples not just the atypical ones.

Other comments/suggestions:
- Metric learning was previously used for adversarial robustness in Metric Learning for Adversarial Robustness, NeurIPS 2019 though in a different way
- In light of the relation to margin, some related work from the margin perspective that could be discussed include
  * Helper-based Adversarial Training: Reducing Excessive Margin to Achieve a Better Accuracy vs. Robustness Trade-off, ICML Workshop on Adversarial ML 2021
  * MMA Training: Direct Input Space Margin Maximization through Adversarial Training, ICLR 2020

  In particular, the first paper achieves similar results as the current work by restricting the margin increase through different means; the downweighting of low (or even negative margin) examples in BAT has a similar effect.

- Typos:
  * Sec 3.2 on p4: trails -> trials
  * Theorem 1: the condition $d_{+1} \leq 2\eta$ should be on $d_{-1}$ instead


**Summary Of The Paper:**

This paper studies the role of atypical samples in the context of adversarial robustness, and further introduces a variant of adversarial training based on these observations. The proposed Benign Adversarial Training (BAT) incorporates an additional temperature-scaled n-pair loss and weights adversarial examples according to their margin. BAT shows improved clean accuracy at similar levels of robustness compared to baselines.

**Summary Of The Review:**

Overall the proposed BAT algorithm achieves promising results that improve clean accuracy at similar robustness levels, but the design of the algorithm seems to involve elements that are orthogonal from the initial motivation and study of atypical examples. The importance of the atypical-ness of an example should be clarified during the discussion period to better motivate the method.

**Post response update:**
The authors have comprehensively addressed my concerns about relation to margin and importance of the atypicality to the components of BAT through ablation studies. I have increased my score accordingly.

---

> ### Author Response · Authors · 2021-11-22
> **Response to Reviewer qCTN**
>
>
> Thank you for reviewing our paper and we are happy to read that "the observations are interesting" and "the analysis is well-executed". In the following, we will clarify your key concerns. Based on the suggestions of the reviewer, we will discuss: **(1)** What is the  differences between "atypical-ness" and margin? **(2)** More clarifications of the theoretical results. **(3)** More clarifications of the proposed method. **(4)** Additional experiments to underline the importance of "atypical samples" to the performance of BAT.
> **(5)** How is the threshold for memorization score $\text{mem}(x)$ is selected ? Here, we first give a formal definition of "margin" (from the papers mentioned by the reviewer) for the discussion in the remaining part: the margin of a sample $x$ on a classification model $f$ is the distance of $x$ to the decision boundary of $f$: $M(x; f) = \min ||\delta||, \text{~~s.t.~} f(x+\delta) \neq f(x).$
> Next, we will answer these questions and provide additional experiments.
>
> **What is the  differences between "atypical-ness" and margin?**
>
> In this paper, the atypical samples are defined as the samples which are far from the main distribution of its labeled class in the data space. For example, in Figure 5 of the paper, we show "atypical plates" in CIFAR100, which are visually very different from the images in the main distribution of "plates".
>
> However, atypical samples do not necessarily have small margins in DNNs. For example, in Figure 1 of the paper, we show that all atypical samples have a large margin, with $l_\infty$ distance to the decision boundary $\geq 8/255$. It is because the WRN28 model can achieve 100\% training adversarial accuracy on CIFAR100. Moreover, in the table below, we approximately calculate the ($l_2$-norm) margin for both typical / atypical samples in the training set of CIFAR100, via the method CW attack [1]. We report the confidence interval of the margins for 500 samples in natural / PGD adversarial training.
>
> | Margin        | Typical         | Atypical        |
> |---------------|-----------------|-----------------|
> | Natural Train | $0.290\pm0.015$ | $0.285\pm0.011$ |
> | Adv. Train    | $0.965\pm0.036$ | $0.935\pm0.027$ |
>
> From the result, we can see that atypical samples do not have significant smaller margins than typical samples for both models. It may be because the strong expressiveness of DNN models to fit data, by giving every training sample a (similarly) high confidence. Moreover, compared to adversarial training, all samples (including typical \& atypical) have small margins. However, it does not suggest every sample is close to the distribution of other classes. These results highlight the difference between "atypical-ness" and "margin".
>
> **Clarification on the theory.**
>
> The reviewer raises concerns about the theory of this paper, by mentioning: one model which trains to accommodate samples near the decision boundary of an optimal classifier (samples with small margins), the model will be less robust. However, it is not true. For example, the method MMA Training~[2] is trained to gradually fit the samples near the current decision boundary. However, MMA Training can effectively improve the model robustness.
>
> Our theory demonstrates: fitting a specific type of atypical samples will hurt the model. They are the samples deviated from their main subpopulation but resemble to the main subpopulation of another class. For example, as shown in Figure 3 in our paper, a "lamp" which is visually very similar to "cup", could hurt the accuracy / robustness of "cup" during adversarial training. Our theory can also well reflect this case, as the atypical sample we define in Theorem 1 \& 2 has a close distance to the wrong class in the data distribution. As we mentioned in Question (1), these "poisoning" atypical samples do not necessarily have smaller margins to the decision boundary in DNNs.
>
> **Clarification on the proposed method.**
>
> Given that the "margin" of a sample does not always align with the "atypicalness" of a sample, our proposed method "BAT" is designed essentially related to "atypical" samples. For example, using $L_{DS}$ loss in Eq.(9) helps the model render compacted representations only for typical samples in one class. It means that the model is not required to output similar representations for those "atypical" samples in this class. In fact, these atypical samples have (visually) very different features to the typical samples in each class.Therefore, our proposed method is also encouraged to output representations which better align with human vision. This is essentially different from the previous paper [3], which applies "metric learning" for all samples in each class (Figure 5 in the paper).
> Second, via downweighting the "atypical" samples as in Eq.(6), we eliminate their poisoning effect on the model performance. However, we still preserve the model's ability to fit typical samples with small confidence.

---

> > ### Author Response · Authors · 2021-11-22
> > **Response to Reviewer qCTN (Part-2)**
> >
> > **Additional Experimental Results**
> >
> > We follow the suggestion of the reviewer, to provide additional experimental results to validate that the atypical samples matters to the performance of BAT. Note that BAT is composed of two main components: (1) Reweight and (2) Discrimination Loss, which both depend on atypical samples. We conduct experiments to show atypical samples matter in both of these two components. We compare: (1) BAT which only reweights atypical samples (Reweight), vs. BAT which reweights all samples (Reweight*). (2) BAT with $L_{DS}$ only regularizing the representations of typical samples (DS), vs. BAT with $L_{DS}$ regularizing the representations of alll samples (DS*). (3) BAT which depends on atypical samples (BAT) vs. BAT which does not depend on atypical samples (BAT*). Note that BAT-Reweight, BAT-Reweight* compared here didn't involve $L_{DS}$; BAT-DS and BAT-DS* didn't involve reweight.
> >
> > |               | Clean Acc. | PGD. Adv. Acc |
> > |--------------:|-----------:|--------------:|
> > |  BAT-Reweight |    60.9    |      27.2     |
> > | BAT-Reweight* |    60.2    |      26.1     |
> > |     BAT-DS    |    61.9    |      27.7     |
> > |    BAT-DS*    |    60.1    |      25.3     |
> > |      BAT      |    62.0    |      28.5     |
> > |      BAT*     |    69.2    |      26.9     |
> >
> > From the results, we could see "atypical samples" matter for both components of BAT. For each version of BAT, once we exclude the factor of atypical samples, we could see the performance drop on either clean accuracy and adversarial robustness.
> >
> > **How is the threshold for memorization score $\text{mem}(x)$ is selected in this paper?**
> >
> > In the analysis of Section 3, to demonstrate the impact of atypical samples in adversarial training, we set atypical samples to have memorization value $>0.15$. In this way, the found atypical samples have sufficient significance to deviate from the main distribution, based on the empirical study in [4]. For the method of BAT, we tune the memorization threshold as a hyperparameter. All the results reported in Section 5 of the paper are with the threshold to be 0.1.
> >
> > **Reference:**
> >
> > [1] Towards Evaluating the Robustness of Neural Networks, Carlini et al, 2017
> >
> > [2] MMA Training: Direct Input Space Margin Maximization through Adversarial Training, Ding et al, 2018
> >
> > [3] Metric Learning for Adversarial Robustness, Mao et al, 2018
> >
> > [4] What neural networks memorize and why: discovering the long tail via influence estimation, Feldman et al, 2020

---

> > > ### Comment · Reviewer_qCTN · 2021-11-29
> > > **Respone to response**
> > >
> > > Thanks for the response and additional experiments. Here are my further comments:
> > >
> > > I'm not sure the analysis in "What is the differences between "atypical-ness" and margin?" is really helping to support the case - the difference between the two in the clean case is > 3 s.d and for the adversarial case > 8 s.d., and I suspect will be even more obviously different (larger mean difference) with higher memorization value (atypicality) thresholds.
> > >
> > > Also, in the response:
> > > > For example, as shown in Figure 3 in our paper, a "lamp" which is visually very similar to "cup", could hurt the accuracy / robustness of "cup" during adversarial training. Our theory can also well reflect this case, as the atypical sample we define in Theorem 1 & 2 has a close distance to the wrong class in the data distribution.
> > >
> > > "close distance to the wrong class" - this seems to me to be the very definition of margin - unless there is something else I am missing here.
> > >
> > > Finally, in the additional experiments, BAT* seems to outperform BAT in clean accuracy (69.2 vs 62.0) - is this correct?

---

> > > > ### Author Response · Authors · 2021-11-29
> > > > **Response to Reviewer**
> > > >
> > > > We thank the reviewer's timely response.
> > > >
> > > > First, we would clarify there is a minor misunderstanding from the reviewer to our result. In our rebuttal about "What is the differences between atypical-ness and margin?", for **natural training**, the margin difference is $0.290 - 0.285 = 0.005$. While, the standard error for Typical \& Atypical samples are $0.015$ and $0.011$ respectively. The difference is only $\approx 0.3\sim0.5$ times of the standard error. Thus, we could not say there is a significant difference between the margins of atypical & atypical samples in Natural training. Similarly, the analysis also holds for **adversarial training**, where the margin difference is $\approx1$ times of standard error.
> > > >
> > > > Second, we'd like to further highlight the difference between margin and atypical-ness here. The small margin refers to those samples with **a small distance (such as Euclidean Distance) defined in input space**,  to the decision boundary or samples from other classes. The (poisoning) atypical samples resembles other classes from the distribution perspective. Intuitively, the atypical samples can have **very similar semantic features to a wrong class**. For example, the "Lamp" in Figure 3 of this paper have very similar shape \& design to the "cups". However, those atypical samples don't necessarily have small distance (i.e., Euclidean Distance) to the wrong classes in the input space.
> > > >
> > > > Moreover, not all atypical samples are close to the distribution of other classes. For example, in Figure 5 in Appendix of this paper, we present several atypical "Plates" (with mem value = 1.0) in CIFAR100. They do not look like either the typical samples in "Plates" (mem value = 0), or any other classes in CIFAR100. These atypical samples also worth studying, as memorizing them can benefit the model's clean accuracy in natural \& adversarial training.
> > > >
> > > > Last, sorry for the typo. It should be 59.2.

---

> > > > > ### Comment · Reviewer_qCTN · 2021-11-30
> > > > > **Thanks for the clarifications**
> > > > >
> > > > > Thanks for the clarifications and I apologize for the silly error. I think you have addressed my main concerns and I will raise the score to 6 accordingly.

---

> > > > > > ### Author Response · Authors · 2021-12-03
> > > > > > **Thanks for the Responses**
> > > > > >
> > > > > > We gratefully appreciate your time in reviewing our paper, your insightful comments and your support! Your positive feedback is very encouraging for us.

---

> ### Author Response · Authors · 2021-11-25
> **About the Rebuttal to Reviewer Response**
>
> Hi, we sincerely appreciate your review. Since we are approaching the end of the discussion period, we would like to enquire if the reviewer has any remaining questions or further points to discuss after our response?

---

### Decision · Program_Chairs · 2022-01-20

**Decision:**

Reject

**Comment:**

This paper tries to improve the training of adversarial deep neural networks by avoiding fitting the “harmful” atypical samples and fitting more “benign” atypical samples.

Overall, the main concerns are

1. The current presentation can easily cause some misunderstandings on the observations made in Section 3, especially [1] and [3] mentioned by the reviewer iXiX.
- The authors may consider moving "related work" to the first half of the submission, and organize existing findings with rare/hard/atypical in a more principle manner.
- Besides, as author mentioned in Section 3.1: "it is equivalent to a classification task based on an extremely small dataset, with one or a few training samples given". Such findings are natural and not novel to the deep learning community. Authors may consider shorten Section 3.1 and elaborate more in Section 3.2.

2. Theorem 1 and 2 do not help much.
- It does not talk about the training algorithm and models, which over simplifies the learning problem.
- Besides, the authors can consider some theoretical results how BAT improve the performance of typical samples, but still preserve the ability to fit those "useful" atypical samples.
This helps to bridge the gap between motivation behind BAT and its algorithm design (raise by reviewer  ytJj and sm19).

3. It is also suggested to make observations more convincing.
- Since authors want to claim their findings are universal, it is better to consider more adv training methods and datasets; it is also better to change the ratio of "normal samples" v.s. "atypical samples". In this way, the effect of atypical samples in adversarial training can be more carefully quantized.